# Rational Neural Networks have Expressivity Advantages

**Maosen Tang** [* 1]   **Alex Townsend** [* 2]

## Abstract

We study neural networks with trainable low-degree rational activation functions and show that they are more expressive and parameter-efficient than modern piecewise-linear and smooth activations such as ELU, LeakyReLU, LogSigmoid, PReLU, ReLU, SELU, CELU, Sigmoid, SiLU, Mish, Softplus, Tanh, Softmin, Softmax, and LogSoftmax. For an error target of $\varepsilon > 0$, we establish approximation-theoretic separations: Any network built from standard fixed activations can be uniformly approximated on compact domains by a rational-activation network with only $\mathrm{poly}(\log\log(1/\varepsilon))$ overhead in size, while the converse provably requires $\Omega(\log(1/\varepsilon))$ parameters in the worst case. This exponential gap persists at the level of full networks and extends to gated activations and transformer-style nonlinearities. In practice, rational activations integrate seamlessly into standard architectures and training pipelines, allowing rationals to match or outperform fixed activations under identical architectures and optimizers.

## 1. Introduction

Deep neural networks owe much of their empirical success to the choice of nonlinear activation functions. Given an affine map $x \mapsto Wx + b$, the activation $\sigma$ determines how features are composed across layers and therefore controls both the expressivity and trainability of the resulting architecture. Earlier generations of deep learning emphasized piecewise-linear functions, most notably ReLU and its variants (LeakyReLU, PReLU, and ReLU6) (Nair & Hinton, 2010; Maas et al., 2013; He et al., 2015; Sandler et al., 2018). More recent architectures, particularly large-scale transformer systems, rely heavily on smooth, fixed activations such as GELU, Swish/SiLU, Mish, and the ELU/SELU/CELU family (Hendrycks & Gimpel, 2016; Ramachandran et al., 2017; Elfwing et al., 2018; Misra, 2020; Clevert et al., 2016; Klambauer et al., 2017; Barron, 2017; Vaswani et al., 2017). In this work, we demonstrate, both theoretically and practically, that trainable low-degree rational activation functions lead to more expressive and efficient neural networks.

The key structural advantage of rational functions is their ability to efficiently represent non-smooth features. Functions with kinks, sharp transitions, or nearby complex singularities (e.g., $|x|$, ReLU, $\sqrt{x}$, or $\log(x)$ near 0) admit exponentially accurate rational approximations but only algebraically accurate approximations by polynomials and many smooth surrogates; this phenomenon is classical in approximation theory (Trefethen, 2019; Gonchar, 2011) and underlies the separation mechanisms we develop at the network level.

A rational neural network replaces the usual scalar activation $\sigma$ with a trainable low-degree rational function, where the coefficients are learned during training. Although each activation is low-degree, composition across layers yields a data-sparse representation of an extremely high-degree rational function (Telgarsky, 2017; Boullé et al., 2020). This paper builds on that viewpoint by proving that rational activations have better expressibility than most modern activation functions, and demonstrating the advantages in supervised vision and offline reinforcement learning.

We prove a separation in parameter complexity between rational networks and networks built from standard smooth activations, including GELU, Swish and SiLU, Mish, ELU, SELU, CELU, Softplus, and Softmax. For every $\varepsilon \in (0, 1)$, any such smooth-activation network on a fixed compact domain can be uniformly approximated by a rational network whose size is bounded by a fixed polynomial in $\log\log(1/\varepsilon)$, and in particular by $\mathcal{O}\big(\log^3\log(1/\varepsilon)\big)$ for the activations we study in detail. In the reverse direction, a worst-case family of rational functions realizable by small rational networks requires any bounded-parameter smooth-activation approximation to uniform error at most $\varepsilon$ to have size at least $\Omega\big(\log(1/\varepsilon)\big)$, so rational networks can be exponentially more expressive than smooth-activation networks in the approximation regime. Empirically, these gains translate

---

[*]Equal contribution  [1]Center for Applied Mathematics, Cornell University, United States [2]Department of Mathematics, Cornell University, United States. Correspondence to: Maosen Tang <mt872@cornell.edu>.

*Proceedings of the 43rd International Conference on Machine Learning*, Seoul, South Korea. PMLR 306, 2026. Copyright 2026 by the author(s).

into improved accuracy and faster convergence in vision and into stronger normalized scores in offline MuJoCo reinforcement learning under fixed architectures and training budgets.

Finally, modern architectures rarely use activations in isolation: normalization layers introduce additional learned affine degrees of freedom and (for batch-based methods) data-dependent stochasticity (Ioffe & Szegedy, 2015; Ba et al., 2016; Wu & He, 2018). For adaptive rational activations, these couplings can create non-identifiability and poor conditioning in the joint parameterization, and can amplify coefficient sensitivities; we formalize these effects in Appendix D and evaluate regimes where reducing or removing normalization is beneficial for training adaptive rationals (Zhu et al., 2025).

**Contributions.**

The main contributions of this paper are:

- We prove size bounds between activation families: rational networks approximate GELU, Swish, and SiLU with size $\mathcal{O}\big(\log^3\log(1/\varepsilon)\big)$ and an $\Omega(\log\log(1/\varepsilon))$ lower bound, while approximating worst-case rational targets by bounded-parameter smooth-activation networks from the same family requires $\Omega(\log(1/\varepsilon))$ size.

- For any function with a large curvature, e.g., $|x|$, ReLU, and $\sqrt{x}$ (on $[0,1]$), we prove that rational neural networks are more expressive than smooth-activation networks.

- We extend these approximation results to gated activations and transformer-style architectures.

- We provide constructive proofs based on classical rational approximation theory, including Zolotarev functions and an arithmetic-geometric-mean construction that demonstrates the expressivity advantage of rational neural networks.

- We show that standard normalization layers can be hurtful for adaptive rational activations.

Our results suggest that rational activations could form a mathematically natural and computationally efficient foundation for deep learning architectures.

## 2. Related Work

Trainable rational activations have been explored and found competitive or superior in reinforcement learning (Delfosse et al., 2024; Surdej et al., 2025), control (Newton & Papachristodoulou, 2023), scientific machine learning and

Green's function learning (Boullé et al., 2022), operator learning (Madala et al., 2025; Wu & Lin, 2025), and transformer architectures (Fang et al., 2023). These empirical studies are supported by practical parameterizations, including open-source Padé-style rational activations (Delfosse et al., 2020; 2024) and the ERA formulation (Trimmel et al., 2022), which improves numerical stability and conditioning.

A separate algebraic and geometric line of work studies polynomial and rational network classes through dimension, identifiability, and neuromanifold structure, including polynomial networks (Kileel et al., 2019; Marchetti et al., 2025; Shahverdi et al., 2025; 2024; Massarenti & Mella, 2025) and rational-function families (Conca et al., 2024; Grosdos et al., 2025). Generalization and covering-number results for real algebraic varieties and rational networks (Zhang & Kileel, 2025) are also relevant. Our results establish quantitative uniform-approximation separations between rational activations and many fixed activation families.

## 3. Rational Neural Networks

We use rational neural networks, meaning standard feedforward networks whose scalar nonlinearity is a trainable rational function. A rational activation is

$$r(x) = \frac{P(x)}{Q(x)} = \frac{\sum_{i=0}^{r_P} a_i x^i}{\sum_{j=0}^{r_Q} b_j x^j}, \tag{1}$$

with trainable coefficients $\{a_i\}_{i=0}^{r_P}$ and $\{b_j\}_{j=0}^{r_Q}$. We call $r_P$ the numerator degree and $r_Q$ the denominator degree, and we require $Q(x) \neq 0$ on the real domain of interest. Classical approximation theory shows that rational functions can approximate targets with sharp transitions far more efficiently than polynomials of comparable degree, see Trefethen (2019) and Gonchar (2011).

A depth $M$ rational network maps an input $x$ to an output $R(x)$ by alternating affine maps and rational activations,

$$
\begin{aligned}
h^{(0)}(x) &= x, \\
z^{(\ell)}(x) &= W^{(\ell)} h^{(\ell-1)}(x) + b^{(\ell)}, \\
h^{(\ell)}(x) &= r^{(\ell)}\big(z^{(\ell)}(x)\big), \qquad \ell = 1, \ldots, M-1, \\
R(x) &= W^{(M)} h^{(M-1)}(x) + b^{(M)},
\end{aligned}
\tag{2}
$$

where each $r^{(\ell)}$ is a rational activation, potentially with its own coefficients. Throughout our theory, the numerator and denominator degrees are uniformly bounded independently of layer index, depth, width, and $\varepsilon$; different layers may learn different coefficients, but the activation family remains fixed. The trainable parameters consist of the affine weights and biases $\{W^{(\ell)}, b^{(\ell)}\}$ together with the rational coefficients in each layer.

A central modeling choice is to use low-degree rationals at every layer and rely on composition across depth. This com-

positional viewpoint matches the role of depth in standard networks and is aligned with rational network approximation theory developed by Telgarsky (2017) and sharpened by Boullé et al. (2020). It also connects to learnable activation work, including Padé activation units Molina et al. (2020), while our emphasis is network-level parameter complexity rather than the degree of a single isolated rational.

In practice, we find that rational neural networks are quite tricky to train because during the learning stage the activation function can develop real poles, possibly causing gradients to explode. Instead, in our experiments, we use a small tweak and use activation functions suggested in Delfosse et al. (2024) together with the accompanying implementation, which enforces a strictly positive denominator via absolute values, so real poles are excluded by construction. Throughout, we use $P = 5$ and $Q = 4$, matching the default in Delfosse et al. (2020).

## 4. Our Theoretical Results

This section compares rational neural networks to networks built from smooth activations, with expressivity measured by the number of trainable parameters required to reach a target uniform error on a fixed compact domain, for example $[-1, 1]^d$. We write $0 < \varepsilon < 1$ for the approximation tolerance and measure error in the $L^\infty$ norm. All constants hidden in $\mathcal{O}[\cdot]$ and $\Omega[\cdot]$ depend only on fixed architectural bounds and are independent of $\varepsilon$.

Our starting point is the established separation between rational and ReLU style nonlinearities in Boullé et al. (2020). For the scalar ReLU map $\text{ReLU}(x) = \max\{x, 0\}$ on $[-1, 1]$, it has been shown that an essentially sharp log–log characterization of rational network approximation: for every $0 < \varepsilon < 1$ there exists a constant-width rational network $R_\varepsilon$ with size $\mathcal{O}(\log \log(1/\varepsilon))$ such that $\|R_\varepsilon - \text{ReLU}\|_{L^\infty([-1,1])} \le \varepsilon$, and conversely any rational network achieving $\|R - \text{ReLU}\|_{L^\infty([-1,1])} \le \varepsilon$ must have size $\Omega(\log \log(1/\varepsilon))$. In the reverse direction, Boullé et al. (2020) also proves that there exist rational networks that cannot be approximated by ReLU networks without a larger logarithmic overhead in size, with a worst-case lower bound of order $\Omega(\log(1/\varepsilon))$ for uniform error $\varepsilon$. This separation extends to a broad ReLU family of piecewise-linear activation functions, including $\text{ReLU}$, $\text{LeakyReLU}$, $\text{PReLU}$, and $\text{ReLU6}$.

At a high level, our results show that rational neural networks can approximate smooth-activation networks with exponentially fewer parameters than the converse. Concretely, we prove that

- The GELU activation can be uniformly approximated on $[-1, 1]$ by a constant-width rational neural network with size $\mathcal{O}(\log^3 \log(1/\varepsilon))$, and any rational network

achieving uniform error at most $\varepsilon$ must have size at least $\Omega(\log \log(1/\varepsilon))$.

- Conversely, there exist simple rational functions that require at least $\Omega(\log(1/\varepsilon))$ parameters to approximate by bounded-parameter GELU networks, while analytic rational functions admit a constructive GELU approximation with size $\mathcal{O}(\log^2(1/\varepsilon))$.

The log-log versus log distinction is substantial in accuracy: a logarithmic parameter law corresponds to errors that decrease exponentially with size, whereas a log-log law corresponds to a doubly exponential decrease up to fixed powers. In the constructions below, the stated size growth is achieved with constant-width networks; the dependence on $\varepsilon$ is carried by depth, and rational degrees stay uniformly bounded.

The same log–log versus log separation extends beyond GELU to a broad family of smooth activations built from transcendental functions, including ELU, LogSigmoid, SELU, CELU, Sigmoid, SiLU, Mish, Softplus, Tanh, Softmin, Softmax, and LogSoftmax, see Sections 4.1 and 4.2. Moreover, these separation results extend to gated activations and transformer architectures, as we discuss in Section 4.4.

### 4.1. Approximation of Smooth Activations by Rational Networks

We begin by showing that rational neural networks can efficiently approximate commonly used smooth activation functions. The key example is the tanh-approximate version of GELU activation

$$G(x) = \frac{x}{2}\big(1 + \tanh(\alpha(x + \beta x^3))\big),$$
$$\alpha = \sqrt{2/\pi}, \ \beta = 0.044715,$$

which is representative of a broad class of smooth, monotone activations used in modern architectures. Throughout this paper, we will we refer to $G(x)$ as GELU.

Our main technical result shows that GELU can be approximated to arbitrary accuracy by a rational neural network of extremely small size.

**Theorem 4.1** (Rational approximation of GELU). *For any* $0 < \varepsilon < 1$*, there exists a rational neural network* $R : [-1, 1] \to [-1, 1]$ *of size*

$$\mathcal{O}\big((\log^3 \log(1/\varepsilon))\big)$$

*such that*

$$\|R - G\|_{L^\infty([-1,1])} \le \varepsilon.$$

*Moreover, no rational neural network of size* $o(\log \log(1/\varepsilon))$ *can achieve this accuracy.*

The proof, given in Appendix A, is constructive and proceeds by decomposing GELU into elementary operations—square roots, logarithms, inverse hyperbolic functions, and tanh—each of which admits a rational neural network realization with doubly exponential convergence under composition. The lower bound follows from classical results in complex rational approximation, which relate the best achievable error to the distance of the nearest complex singularities of the target function.

The same reasoning extends from GELU to the broader transcendental activation family ELU, LogSigmoid, SELU, CELU, Sigmoid, SiLU, Mish, Softplus, Tanh, Softmin, Softmax, and LogSoftmax. For the upper bound, each of these activations can be written as a fixed finite composition of tanh, exp, log, and affine maps, so substituting the corresponding rational subnetworks from Appendix A yields the same $\mathcal{O}\big(\log^3\log(1/\varepsilon)\big)$ size bound up to constants. For the lower bound, these activations likewise have complex singularities at constant distance from the real interval, so the same singularity-driven obstruction yields an $\Omega\big(\log\log(1/\varepsilon)\big)$ size lower bound by the Appendix A argument.

### 4.2. Approximation of Rational Functions by Smooth Activations

We now turn to the converse question: how efficiently can smooth-activation networks approximate rational functions?

Although networks with transcendental smooth activations such as GELU, Swish, and SiLU are universal approximators, bounded-parameter models from this class face a quantitative barrier. When weights and biases are uniformly bounded, the network derivatives are correspondingly controlled, so matching the large curvature required to approximate rational targets with nearby poles forces the parameter count to grow at least logarithmically in $1/\varepsilon$.

**Theorem 4.2** (Lower bound for GELU approximation of rationals). *For every sufficiently small $0 < \varepsilon < 1$, there exists a rational function $R_\varepsilon : [-1, 1] \to \mathbb{R}$ such that any scalar-input GELU network $F$ with uniformly bounded parameters satisfying*

$$\|F - R_\varepsilon\|_{L^\infty([-1,1])} \leq \varepsilon,$$

*the size of $F$ must be at least $\Omega(\log(1/\varepsilon))$.*

The proof, given in Appendix B, exploits curvature constraints for a worst-case family of $R_\varepsilon$ with poles near the real line. Many functions achieve a similar lower bound, and one can immediately witness a log versus log-log separation between GELU and rational neural networks by considering functions such as $|x|$ on $[-1, 1]$.

We complement this lower bound with a constructive upper bound.

**Theorem 4.3** (GELU approximation of rational functions). *Let $R : [-1, 1] \to [-1, 1]$ be a rational function analytic in a neighborhood of $[-1, 1]$. Then, for any $0 < \varepsilon < 1$, there exists a GELU network $F$ of size*

$$\mathcal{O}\big(\log^2(1/\varepsilon)\big)$$

*such that*

$$\|F - R\|_{L^\infty([-1,1])} \leq \varepsilon.$$

The construction is based on Chebyshev expansions and an inexact Clenshaw recurrence, implemented using carefully designed GELU-based multiplication and squaring blocks.

The GELU proofs in Appendix B are representative of the full transcendental activation family discussed above. For the upper bound, the Chebyshev approximation plus inexact Clenshaw evaluation only requires that the activation gives a way to achieve a small-argument squaring and multiplication operator from a local Taylor expansion, so the same $\mathcal{O}\big(\log^2(1/\varepsilon)\big)$ construction applies to any such smooth activation when the rational target is analytic in an open neighborhood of $[-1, 1]$. For the lower bound, the hard instances are rational functions with a complex pole near $[-1, 1]$, which forces any uniform $\varepsilon$-approximant to exhibit second derivatives of order $\varepsilon^{-1}$ somewhere on the interval. Under bounded parameters, smooth-activation networks have uniformly controlled derivative growth, so achieving this curvature necessarily incurs the same worst-case $\Omega\big(\log(1/\varepsilon)\big)$ size barrier beyond GELU, exactly as in Appendix B.

Together, Theorems 4.1 to 4.3 establish a strict expressive asymmetry between rational and smooth-activation networks.

### 4.3. Lifting Scalar Results to Full Networks

The preceding results concern scalar functions. We now lift them to full feedforward architectures acting on $[-1, 1]^d$.

**Theorem 4.4** (Network-level approximation equivalence). *Let $f : [-1, 1]^d \to [-1, 1]$ be a neural network with $M$ layers and at most $k$ nodes per layer.*

1. *If $f$ uses GELU (or Swish or SiLU) activations, then for any $0 < \varepsilon < 1$ there exists a rational neural network $R$ of size*

   $$\mathcal{O}\big(kM \log^3 \log(1/\varepsilon)\big)$$

   *such that $\|f - R\|_\infty \leq \varepsilon$.*

2. *If $f$ is a rational neural network, then there are worst-case rational neural network targets for which any GELU network approximating $f$ to accuracy $\varepsilon$ must have size at least $\Omega(kM \log(1/\varepsilon))$.*

The proof propagates scalar approximation errors layer by layer using Lipschitz bounds on the activations and standard stability arguments. Details are provided in Appendix C.

### 4.4. Extensions to Gated Activations and Transformers

For gated activations, gated rational and gated smooth models share the same gating wrapper and differ only in the scalar activation used inside the gated block. In the smooth-to-rational direction, bounded parameters keep every activation input in a compact range, so replacing each smooth activation call by a uniformly accurate rational surrogate yields a gated rational model that emulates the gated smooth model with overhead equal to the scalar replacement cost. In the rational-to-smooth direction, fix $0 < \varepsilon < \frac{1}{8}$ and set $\eta = \sqrt{\varepsilon}$. We choose a gated rational target whose overall scalar input-output map is $r_\eta(x) = \frac{\eta^2}{x^2+\eta^2}, x \in [-1, 1]$. This target is uniformly bounded by 1, but it satisfies $r_\eta''(0) = -2/\eta^2 = -2/\varepsilon$. Therefore, any bounded-parameter gated smooth model that approximates this target uniformly to error $\varepsilon$ gives, as an input-output map, a smooth approximation to $r_\eta$ with the same accuracy. The finite-difference argument in Appendix B then forces curvature of order $\varepsilon^{-1}$, and the bounded-parameter derivative-growth bound gives the same $\Omega(\log(1/\varepsilon))$ size lower bound on the smooth side. Therefore, the same log-log versus log separation gap persists between gated rational networks and gated smooth-activation networks.

For transformers, we keep attention, normalization, and residual wiring unchanged and only replace the activation used inside the MLP sublayer. Since the overall architecture is identical outside the MLP, the representational difference between the two transformer families is inherited from the pointwise nonlinearity inside each MLP block, so the scalar separation implies an expressivity advantage for transformers with rational MLP activations at a comparable parameter budget. Many modern transformers use gated MLPs, and the same conclusion applies in that setting by the gated extension established above, so swapping the MLP activation to a rational one is expected to preserve the advantage while leaving the rest of the transformer block untouched.

### 4.5. Normalization and Adaptive Rational Activations

Our separation results are approximation-theoretic, but in modern training pipelines the activation is typically coupled to normalization layers. For adaptive rational activations, this coupling is intrinsically disadvantageous. First, normalization applies a learned affine transform, and for rationals this affine freedom can be absorbed by reparameterizing the rational coefficients, creating non-identifiability and poor conditioning in the joint parameterization. This interaction arises under standard normalization operators, including batch normalization (Ioffe & Szegedy, 2015), layer normalization (Ba et al., 2016), and group normalization (Wu & He, 2018), and it is specific to adaptive rational activations because the activation itself carries trainable degrees of freedom. Second, when normalization uses data-dependent statistics (most notably during batch normalization), the induced stochasticity in normalized pre-activations is amplified by the polynomial coefficient sensitivities inherent to trainable rationals, increasing gradient noise and making optimization more brittle. Appendix D formalizes these mechanisms for a "safe" Padé-like parameterization and makes explicit the resulting invariances and coefficient-gradient bounds. Consequently, normalization is not a neutral architectural component for adaptive rationals: it can suppress the benefits of adaptivity and can directly hinder stable training.

In transformer architectures, this issue is particularly acute because LayerNorm is applied pervasively and its affine parameters are trained jointly with the MLP nonlinearity. The preceding mechanisms therefore predict stronger activation–normalization interference when the nonlinearity itself is adaptive. Motivated by recent evidence that transformers can be trained successfully without normalization (Zhu et al., 2025), we also consider "no-LayerNorm" transformer variants. In this regime, the adaptive activation provides an additional source of learnable rescaling and shaping, allowing the model to compensate for the removal of LayerNorm while retaining stable training. Empirically, this combination yields strong performance, consistent with the view that removing normalization can eliminate a major source of conditioning pathology for adaptive rational activations while preserving the representational benefits of adaptivity.

### 4.6. Implications

Taken together, these results establish a sharp and robust separation between rational neural networks and smooth-activation networks. While smooth activations such as GELU are sufficient for universal approximation, rational activations achieve the same expressive power with exponentially fewer parameters in regimes involving non-smooth structure or nearby complex singularities.

From a theoretical standpoint, this explains why rational neural networks consistently outperform smooth activations in practice on problems with sharp transitions or multiscale structure. From a modeling standpoint, it suggests that rational activations provide a principled and efficient foundation for modern deep learning architectures.

## 5. Experiments

We evaluate the effect of replacing fixed activation functions with trainable rational activations in supervised vision and offline continuous-control reinforcement learning. Across all settings, we keep architectures, optimizers, training budgets, and data fixed within each experimental block, and varying only the activation.

*Table 1.* CIFAR-10 top-1 test accuracy. Plain VGG without additional accuracy boosters. Mean ± standard deviation across five seeds. Best and second-best within each model block are highlighted.

| Model | Act | Top-1 Acc ↑ | Time ↓ |
|---|---|---|---|
| | GELU | 79.93 ± 0.72 | 50 ± 0 |
| | ReLU | 79.03 ± 0.27 | 50 ± 0 |
| VGG4 | Swish | 80.16 ± 0.28 | 50 ± 0 |
| | LeakyReLU | 79.19 ± 0.44 | 50 ± 0 |
| | **Rational** | **81.96 ± 0.33** | 52 ± 0 |
| | GELU | 80.48 ± 0.58 | 54 ± 0 |
| | ReLU | 82.00 ± 0.27 | 54 ± 0 |
| VGG8 | Swish | 78.71 ± 0.72 | 54 ± 0 |
| | LeakyReLU | 82.19 ± 0.21 | 54 ± 0 |
| | **Rational** | **84.89 ± 0.19** | 58 ± 0 |

## 5.1. CIFAR-10 Image Classification

We evaluate rational activations on CIFAR-10 (Krizhevsky & Hinton, 2009) using VGG4 and VGG8, keeping the architecture family fixed and changing only the activation. We report top-1 test accuracy as the best value over training, with mean ± standard deviation. Table 1 reports the baseline comparison. Table 2 augments training with label smoothing, Mixup, dropout before the classifier, and weight decay, and it additionally evaluates each activation with and without GroupNorm.

The baseline VGG results isolate the effect of the activation by removing auxiliary training techniques and keeping the architecture and optimizer fixed. In this setting, rational activations achieve the strongest accuracy in both VGG4 and VGG8 in Table 1, and VGG4 with a rational activation reaches the same accuracy range as VGG8 equipped with the best fixed activations, reflecting the parameter-efficiency predicted by our approximation-theoretic separations.

Under the augmented pipeline, Table 2 reports, for each activation and each choice of using GroupNorm, the better of exactly two initialization settings, the default initialization and the LSUV toggle. For rational activations we additionally compare two starting shapes, with Rational 1 initialized to behave like LeakyReLU and Rational 2 initialized to behave like GELU. Without GroupNorm, the rational variants are uniformly best for both VGG4 and VGG8. With GroupNorm, fixed activations either change only slightly or improve markedly, while their occasional degradations remain modest; in contrast, GroupNorm consistently induces a much larger drop for the rational variants. Despite this drop, rationals remain top or near-top under GroupNorm, retaining best or second-best performance within each model block, and the two rational initializations separate more with behavior that reflects properties inherited from the corresponding fixed activations, indicating that a suitable initialization helps rationals remain competitive across training regimes.

*Table 2.* CIFAR-10 top-1 test accuracy under a strengthened training recipe. Mean ± standard deviation across five seeds. For each activation we bold the better result between the no GroupNorm and GroupNorm variants. Best and second-best activations within each model are highlighted.

| Model | Act | no GN Top-1 Acc ↑ | GN Top-1 Acc ↑ | Time ↓ |
|---|---|---|---|---|
| | GELU | **89.09 ± 0.10** | 89.06 ± 0.35 | 50 ± 0 |
| | ReLU | **89.21 ± 0.13** | 88.30 ± 0.73 | 50 ± 0 |
| VGG4 | Swish | 88.33 ± 0.15 | **88.59 ± 0.24** | 50 ± 0 |
| | LeakyReLU | **89.03 ± 0.18** | 88.16 ± 0.53 | 50 ± 0 |
| | Rational 1 | **90.62 ± 0.13** | 88.11 ± 0.37 | 52 ± 0 |
| | Rational 2 | **90.67 ± 0.17** | 89.06 ± 0.64 | 52 ± 0 |
| | GELU | 91.64 ± 0.04 | **92.21 ± 0.24** | 54 ± 0 |
| | ReLU | **91.73 ± 0.11** | 91.61 ± 0.29 | 54 ± 0 |
| VGG8 | Swish | 90.96 ± 0.12 | **92.09 ± 0.24** | 54 ± 0 |
| | LeakyReLU | **91.65 ± 0.28** | 91.64 ± 0.12 | 54 ± 0 |
| | Rational 1 | **92.57 ± 0.15** | 91.80 ± 0.31 | 58 ± 0 |
| | Rational 2 | **92.50 ± 0.19** | 92.13 ± 0.31 | 58 ± 0 |

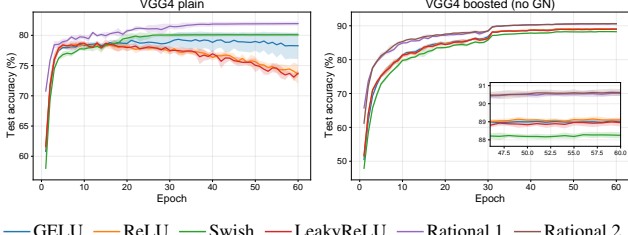

*Figure 1.* CIFAR-10 test-accuracy score curves for VGG4 in the plain setting (left) and the boosted setting without GroupNorm (right). Curves show mean across five seeds with ±1 standard-deviation shading. Rational converges earlier and to a better final accuracy than the fixed activations in both settings.

Figure 1 visualizes the test-accuracy score curves for VGG4 in the baseline setting and in the augmented pipeline without GroupNorm. In both cases, rational activations reach strong accuracy earlier in training and converge to a better final value than the fixed baselines, so the score curves corroborate the advantages observed in the best-over-training summaries. Additional score curves for the remaining model and training variants are provided in Appendix E.

Beyond scalar accuracy, we directly inspect the learned shape of the Rational nonlinearity. Across most layers in the VGG8 feature stack, the activation starts close to GELU and quickly becomes uneven, with the curve dropping into a small dip near zero and then rising into a bump on the positive side, and these changes sit where the layer inputs appear most often in the histogram overlay. Figure 2 shows a representative layer, and the full layerwise evolution across depth is provided in Appendix E. This kind of learned shape can be favorable: smooth activations with built in gating have repeatedly performed well in modern image models, and non monotone examples such as Swish and Mish report improved optimization and accuracy in practice (Ramachandran et al., 2017; Misra, 2020). If one closely examines Figure 2, there's a curiosity: the hidden-

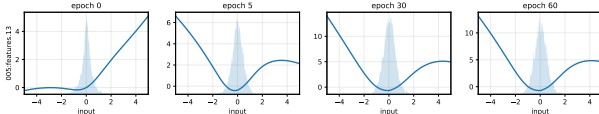

*Figure 2.* Example learned Rational shape in VGG8 (representative feature-layer snapshot). Curves show the learned scalar nonlinearity at epochs 0, 5, 30, and 60; the light shaded overlay is the empirical density of the corresponding layer's pre-activation inputs estimated from held-out mini-batches and plotted on a secondary axis. The Rational quickly reallocates curvature to the high-density input region and develops a localized non-monotone gating shape.

layer rational activations are evaluated mostly in regions where the rationals are smooth and slowly varying, so they are not being used to approximate kinks or high curvature; this suggests that an alternative rational architecture might better exploit rational expressivity and achieve substantially higher scores.

### 5.2. Offline Reinforcement Learning

We study offline reinforcement learning on continuous-control locomotion benchmarks in MuJoCo (Todorov et al., 2012) using the Minari *medium* datasets (Younis et al., 2024), where training is restricted to a fixed collection of logged transitions with no additional environment interaction. Following the D4RL locomotion convention, the medium datasets are generated by a partially trained behavior policy, so they contain sustained locomotion and nontrivial rewards, yet they do not provide expert-level coverage of the highest-return state–action regions (Fu et al., 2021).

We evaluate learned policies by rollouts in the corresponding Gymnasium v5 environments (Towers et al., 2024) and report both raw episode return and a D4RL-style normalized score (Fu et al., 2021) computed from random and expert anchors re-estimated to match this Gymnasium v5 evaluation specification, using the corresponding Minari expert dataset for the expert anchor. This normalization places HalfCheetah, Hopper, and Walker2d on a shared scale and supports cross-task comparisons between activations.

We compare Implicit Q-Learning (Kostrikov et al., 2021) and TD3+BC (Fujimoto & Gu, 2021) on the same tasks while keeping the algorithm and network architecture fixed and changing only the activation function. We sweep ReLU, SiLU, GELU, and Rational. Table 3 reports the final performance summaries for both algorithms in a single comparison table. Across tasks and algorithms, rational activations surpass or match the strongest fixed activation in five of six settings under normalized score. Moreover, when rational leads, it typically does so by a clear margin, while in the single setting where it does not lead, the gap to the best fixed activation is small. This pattern is consistent with

the parameter-efficiency motivation of our work: with the same network width and depth, a rational nonlinearity can efficiently realize the effective shapes needed by the best fixed activations, whereas a fixed activation typically needs additional architectural overhead to reproduce behaviors that a low-degree rational can represent within the same parameter budget.

To probe parameter efficiency more directly, we additionally ran an IQL width sweep with smaller and larger two-layer MLPs. The original setting uses width 256; the smaller and larger settings use widths 128 and 384, respectively. Rational models add only 80 trainable activation parameters relative to matched GELU models. Table 4 reports five-seed means and standard deviations for normalized score and return, and shows that the smaller rational model is competitive with, and in some tasks substantially better than, much larger GELU models, except where the task is already near saturation.

Figure 3 summarizes two representative offline RL diagnostics. The top panels highlight learning curves for IQL on HalfCheetah-medium and TD3+BC on Walker2d-medium, showing v5-normalized evaluation score versus gradient updates under each activation. Rational reaches strong performance earlier in training and attains a higher plateau than the fixed activations in these examples, so the curves corroborate the best-over-training summaries in Table 3. Full learning curves across tasks and both algorithms are provided in Appendix F.

The bottom panel inspects a representative Rational layer from the IQL HalfCheetah actor network. The learned curve quickly departs from its initialization target and reallocates curvature toward the high-density input region, yielding a localized gating shape that is aligned with the statistics induced by the fixed dataset and the evolving policy.

### 5.3. Tiny ImageNet Image Classification

We evaluate rational activations in transformer-based image classifiers on Tiny ImageNet, using ViT S 8, CaiT S24, and Swin T, while keeping each model family fixed and changing only the activation function (Vaswani et al., 2017; Dosovitskiy et al., 2021; Touvron et al., 2021; Liu et al., 2021). We report top-1 validation accuracy using an exponential moving average (EMA) of the model weights, meaning we maintain a smoothed copy of the weights throughout training by updating it as a weighted average of the previous EMA weights and the current weights, and we evaluate accuracy with this EMA copy at the best checkpoint; we also report total training time.

Table 5 shows that swapping the standard activation for a trainable low-degree rational nonlinearity typically matches or improves the best EMA validation accuracy within each

*Table 3.* Offline RL on MuJoCo medium datasets. Mean ± standard deviation across five seeds. Best and second-best v5 normalized scores within each task block are highlighted. Left block: IQL. Right block: TD3+BC.

| | | **IQL** | | |
|---|---|---|---|---|
| Task | Act | Norm score ↑ | Return ↑ | Time ↓ |
| HalfCheetah | GELU | 79.32 ± 7.04 | 12823.84 ± 1164.38 | 53 ± 1 |
| | ReLU | 85.78 ± 6.35 | 13891.68 ± 1049.75 | 55 ± 1 |
| | SiLU | 77.46 ± 7.93 | 12515.82 ± 1310.57 | 54 ± 1 |
| | **Rational** | **94.71 ± 0.78** | **15368.13 ± 129.22** | 71 ± 2 |
| Hopper | GELU | 90.75 ± 1.60 | 3502.30 ± 61.32 | 54 ± 1 |
| | ReLU | 90.69 ± 1.60 | 3499.87 ± 61.63 | 55 ± 1 |
| | SiLU | 89.64 ± 2.38 | 3459.64 ± 91.53 | 57 ± 1 |
| | **Rational** | **91.58 ± 0.66** | **3534.30 ± 25.55** | 73 ± 1 |
| Walker2d | GELU | 54.95 ± 10.77 | 3763.94 ± 737.55 | 56 ± 2 |
| | ReLU | 61.25 ± 6.46 | 4195.10 ± 442.12 | 54 ± 1 |
| | SiLU | 58.15 ± 5.74 | 3983.11 ± 393.17 | 55 ± 0 |
| | **Rational** | **72.28 ± 11.52** | **4950.47 ± 788.80** | 77 ± 2 |

| | | **TD3+BC** | | |
|---|---|---|---|---|
| Task | Act | Norm score ↑ | Return ↑ | Time ↓ |
| HalfCheetah | **GELU** | **62.61 ± 5.00** | **10059.83 ± 826.69** | 34 ± 0 |
| | ReLU | 58.82 ± 5.05 | 9433.06 ± 835.50 | 35 ± 2 |
| | SiLU | 53.55 ± 5.18 | 8560.98 ± 857.22 | 34 ± 0 |
| | Rational | 62.44 ± 6.11 | 10031.73 ± 1010.89 | 47 ± 1 |
| Hopper | GELU | 92.43 ± 0.41 | 3567.01 ± 15.90 | 34 ± 0 |
| | ReLU | 92.56 ± 0.37 | 3572.08 ± 14.07 | 35 ± 0 |
| | SiLU | 90.90 ± 2.48 | 3508.26 ± 95.12 | 35 ± 0 |
| | **Rational** | **93.05 ± 0.19** | **3590.53 ± 7.16** | 48 ± 1 |
| Walker2d | GELU | 65.23 ± 8.62 | 4467.87 ± 589.96 | 34 ± 1 |
| | ReLU | 69.02 ± 13.01 | 4727.24 ± 890.54 | 34 ± 1 |
| | SiLU | 60.42 ± 9.99 | 4138.02 ± 684.09 | 34 ± 1 |
| | **Rational** | **84.65 ± 1.82** | **5796.67 ± 124.49** | 46 ± 1 |

*Table 4.* IQL width sweep on MuJoCo medium datasets. Mean ± standard deviation across five seeds.

| Task | Width / Params | Act | Norm score ↑ | Return ↑ |
|---|---|---|---|---|
| HalfCheetah | 128 / 77,967 | GELU | 66.11 ± 11.36 | 10638.87 ± 1879.24 |
| | 128 / 78,047 | **Rational** | 92.53 ± 1.47 | 15006.96 ± 242.66 |
| | 384 / 627,087 | GELU | 91.00 ± 3.41 | 14754.40 ± 564.32 |
| | 384 / 627,167 | **Rational** | **94.58 ± 1.08** | **15346.15 ± 177.81** |
| Hopper | 128 / 73,737 | GELU | 91.10 ± 1.26 | 3515.93 ± 48.23 |
| | 128 / 73,817 | **Rational** | 91.70 ± 1.01 | 3538.75 ± 39.00 |
| | 384 / 614,409 | GELU | 91.56 ± 1.07 | 3533.54 ± 40.99 |
| | 384 / 614,489 | **Rational** | **92.39 ± 0.71** | **3565.42 ± 27.29** |
| Walker2d | 128 / 77,967 | GELU | 51.48 ± 7.34 | 3526.54 ± 502.42 |
| | 128 / 78,047 | **Rational** | 60.01 ± 13.06 | 4110.27 ± 894.15 |
| | 384 / 627,087 | GELU | 66.45 ± 10.86 | 4551.08 ± 743.72 |
| | 384 / 627,167 | **Rational** | **78.62 ± 3.86** | **5383.99 ± 263.95** |

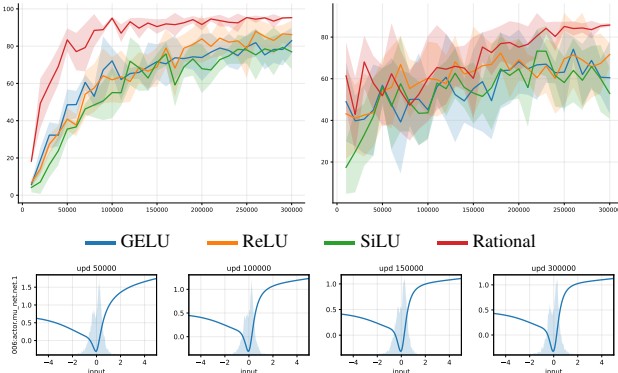

*Figure 3.* Offline RL diagnostics. Top: learning curves under Gymnasium v5 evaluation for IQL on HalfCheetah-medium (left) and TD3+BC on Walker2d-medium (right); lines show mean across five seeds with one standard deviation shading. Bottom: representative learned Rational shape from the IQL actor on HalfCheetah-medium, with the light shaded overlay showing the empirical density of corresponding layer pre-activation inputs estimated from offline mini-batches. Appendix F provides broader score curves and layerwise snapshots.

transformer family, with modest runtime overhead. We additionally test a normalization-free ViT variant motivated by recent evidence that transformer normalization can be removed when paired with an appropriate adaptive element-wise nonlinearity (Zhu et al., 2025) . In this modified model, the rational activation attains a large gain on Tiny ImageNet, while the fixed-activation variants diverge to NaNs and produce no usable checkpoint under the same setup, aligning with our broader finding that normalization can be disproportionately restrictive for adaptive rational nonlinearities.

Figure 4 plots the full EMA accuracy trajectories: for CaiT-S24 the rational run separates clearly and attains a higher plateau, for Swin-T the GELU and rational curves remain very close and repeatedly trade the lead over training, and for ViT-Small the rational run improves over GELU, with the normalization-free ViT-Small variant converging earlier and reaching its plateau sooner, allowing a smaller ViT-Small model with 21.7M parameters to achieve EMA accuracy comparable to Swin-T with 28.3M parameters.

When the structure of the transformer permits, removing the LayerNorm for rational neural networks can yield a significant score boost. However, in some cases, we find

that removing the LayerNorm is detrimental to training. Importantly, even in cases when we cannot remove the LayerNorm, we find that we match the scores attained by the standard activation functions.

## 6. Limitations, weaknesses and future work

Our empirical evaluation is constrained by the scale and breadth of experiments we can run, so our conclusions are grounded in small-to-mid scale vision models and offline MuJoCo benchmarks under fixed architectures and training budgets. All experiments were run on a single RTX 4070 laptop, which prevents us from studying the largest training regimes and from running the widest seed and hyperparameter sweeps. In particular, we do not yet evaluate LLM-scale training, larger and more diverse reinforcement learning suites, or a wide range of transformer variants and training

*Table 5.* Tiny ImageNet top-1 EMA validation accuracy at the best checkpoint. Total time reports the full training time in minutes. Best and second best within each model block are highlighted. In the no-norm ViT rows, "NaN" means that the run diverged and produced no usable checkpoint under the shared setup.

| Model | Act | EMA Val Acc ↑ | Time min ↓ |
|---|---|---|---|
| CaiT S24 | ReLU | 52.94 | 481 |
| | GELU | 55.30 | 465 |
| | **Rational** | **56.15** | 526 |
| Swin T | ReLU | 60.30 | 372 |
| | **GELU** | **60.76** | 372 |
| | Rational | 60.66 | 396 |
| ViT S 8 | ReLU | 57.26 | 600 |
| | GELU | 58.15 | 591 |
| | **Rational** | **58.35** | 637 |
| ViT S 8 no norm | ReLU | *NaN* | *NaN* |
| | GELU | *NaN* | *NaN* |
| | **Rational** | **61.24** | 611 |

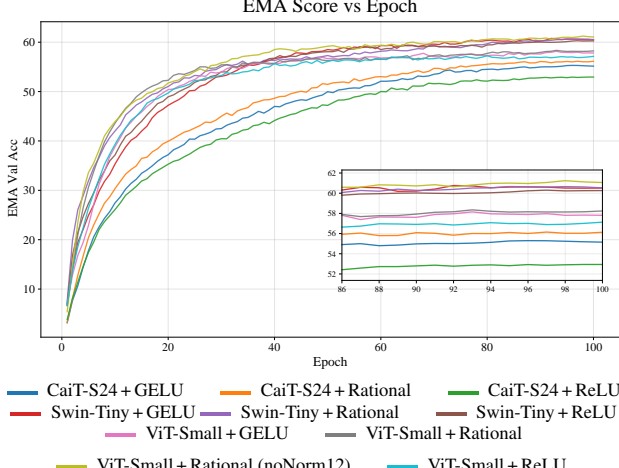

EMA Score vs Epoch

Legend:
— CaiT-S24 + GELU   — CaiT-S24 + Rational   — CaiT-S24 + ReLU
— Swin-Tiny + GELU   — Swin-Tiny + Rational   — Swin-Tiny + ReLU
— ViT-Small + GELU   — ViT-Small + Rational
— ViT-Small + Rational (noNorm12)   — ViT-Small + ReLU

*Figure 4.* EMA validation top-1 accuracy versus epoch on Tiny ImageNet for CaiT-S24, Swin-Tiny, and ViT-Small under different activations. The inset zooms into the final epochs to highlight late-training differences.

recipes within ViT, CaiT, and Swin; broader sweeps and stronger variance studies would better characterize how the observed gains change with scale.

Our wall-clock timings are also influenced by implementation details. We rely on an existing rational-activation implementation rather than an in-house fused PyTorch kernel, so a portion of the runtime cost reflects framework and kernel-launch overhead that could be reduced with a native implementation. Looking forward, we aim to scale these evaluations to large language models and larger-scale reinforcement learning, where the approximation-theoretic motivation and the earlier, often better convergence observed here suggest rational activations may be especially effective.

## Impact Statement

This paper presents work whose goal is to advance the field of Machine Learning. There are many potential societal consequences of our work, none of which we feel must be specifically highlighted here.

## Acknowledgements

This work was supported by the Defense Advanced Research Projects Agency (DARPA) through The Right Space (TRS) Disruption Opportunity (DARPA-PA-24-04-07). The views, findings, and conclusions expressed in this paper are those of the authors and do not necessarily reflect the official policy or position of DARPA, the U.S. Department of Defense, or the U.S. Government.

## Software and Data

All experiments used a single NVIDIA RTX 4070 laptop GPU with Python 3.13.3 and PyTorch 2.8.0. Code is available at https://github.com/mtang398/Rational-Neural-Network. CIFAR-10, Tiny ImageNet, and the Minari MuJoCo medium datasets are publicly available through their standard open-source distributions.

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

# A. Rational Neural Network Approximation of the GELU Activation

This appendix shows that the GELU activation can be approximated on $[-1, 1]$ to within any $\varepsilon > 0$ by a rational neural network of size $\mathcal{O}((\log \log(1/\varepsilon))^3)$. Moreover, we show that it is impossible to achieve this with a rational neural network of size smaller than $\Omega(\log \log(1/\varepsilon))$.

## A.1. The construction of a rational neural network that approximates GELU

Let $\varepsilon > 0$. Our goal in this section is to construct a rational neural network that approximates the GELU activation

$$G(x) = \tfrac{x}{2}\big(1 + \tanh(\alpha(x + \beta x^3))\big), \qquad \alpha = \sqrt{2/\pi}, \ \beta = 0.044715$$

to within $\varepsilon$ in the uniform norm on $[-1, 1]$. The argument proceeds in five steps: (1) We build a constant-width rational neural network $R_{\mathrm{sqrt}}$ that approximates $\sqrt{x}$ on $[0, 1]$ of depth $\mathcal{O}(\log \log(1/\varepsilon))$, (2) We combine $R_{\mathrm{sqrt}}$ with the Sasaki–Kanada AGM–theta representation of $\log$ (Sasaki & Kanada, 1982; Borwein & Borwein, 1987) to obtain a logarithm rational neural network $R_{\mathrm{log}}$ on an interval with depth $\mathcal{O}((\log \log(1/\varepsilon))^2)$, (3) We express $\operatorname{artanh}(z) = \tfrac{1}{2}(\log(1 + z) - \log(1 - z))$ and use $R_{\mathrm{log}}$ together with the identities $\log(ab) = \log(a) + \log(b)$ and $\log(1) = 0$ to obtain an $\operatorname{artanh}$ block $R_{\mathrm{artanh}}$ on a small interval $[-1/8, 1/8]$, again with size $\mathcal{O}((\log \log(1/\varepsilon))^2)$. Fourth, we apply Halley's method to the equation $\operatorname{artanh}(t) = s$, substituting $R_{\mathrm{artanh}}$ for $\operatorname{artanh}$, to construct a $\tanh$ block $R_{\mathrm{tanh}}$ on $[-1/8, 1/8]$ whose size scales like $\mathcal{O}((\log \log(1/\varepsilon))^3)$. Finally, we compose $R_{\mathrm{tanh}}$ with the fixed cubic polynomial $P(x) = \alpha(x + \beta x^3)$ and a finite double-angle ladder for $\tanh$ to rescale $P(x)$ into $[-1/8, 1/8]$, obtaining a constant-width rational network $R_{\mathrm{GELU}}$ that uniformly approximates $G(x)$ on $[-1, 1]$ to within $\varepsilon$ with total size $\mathcal{O}((\log \log(1/\varepsilon))^3)$. The convergence behavior of the core steps used in this construction is illustrated in Figure 5.

STEP 1: APPROXIMATING $x^{1/p}$ ON A FIXED INTERVAL

We fix an integer $p \geq 2$ and a parameter $\alpha_0 \in (0, 1)$, and we work on the interval

$$x \in [\alpha_0^p, 1].$$

Following the scalar $p$th–root iteration of Gawlik & Nakatsukasa (2021, Sec. 2), define for $\alpha \in (0, 1)$

$$\mu(\alpha) := \left(\frac{\alpha - \alpha^p}{(p-1)(1-\alpha)}\right)^{1/p}, \tag{3}$$

and consider the coupled recursion

$$
\begin{aligned}
f_0(x) &= 1, & \alpha_0 &= \alpha_0, \\
f_{k+1}(x) &= \frac{1}{p}\left((p-1)\,\mu(\alpha_k)\,f_k(x) + \frac{x}{\mu(\alpha_k)^{p-1} f_k(x)^{p-1}}\right), & k &\geq 0, \\
\alpha_{k+1} &= \frac{p\,\alpha_k}{(p-1)\,\mu(\alpha_k) + \mu(\alpha_k)^{1-p}\alpha_k^p}, & k &\geq 0.
\end{aligned}
\tag{4}
$$

As in Gawlik & Nakatsukasa (2021, Sec. 2), we introduce the scaled approximants

$$\tilde{f}_k(x) := \frac{2\alpha_k}{1 + \alpha_k}\,f_k(x), \qquad k \geq 0. \tag{5}$$

For each fixed $k$, the map $x \mapsto \tilde{f}_k(x)$ is a rational function obtained by composing the fixed bivariate rational update

$$(x, f_k, \alpha_k) \longmapsto (f_{k+1}, \alpha_{k+1})$$

a total of $k$ times, starting from $(f_0, \alpha_0) = (1, \alpha_0)$.

**Lemma A.1** (Composite $p$th root on a fixed interval). *Fix $p \geq 2$ and $\alpha_0 \in (0, 1)$, and let $\tilde{f}_k$ be defined by Equation (4)–Equation (5). Then there exist constants $C_p, c_p > 0$, depending only on $p$ and $\alpha_0$, such that*

$$\sup_{x \in [\alpha_0^p, 1]} \left| \frac{\tilde{f}_k(x) - x^{1/p}}{x^{1/p}} \right| \leq C_p \exp\!\big(-c_p\, 2^k\big) \qquad \text{for all } k \geq 0. \tag{6}$$

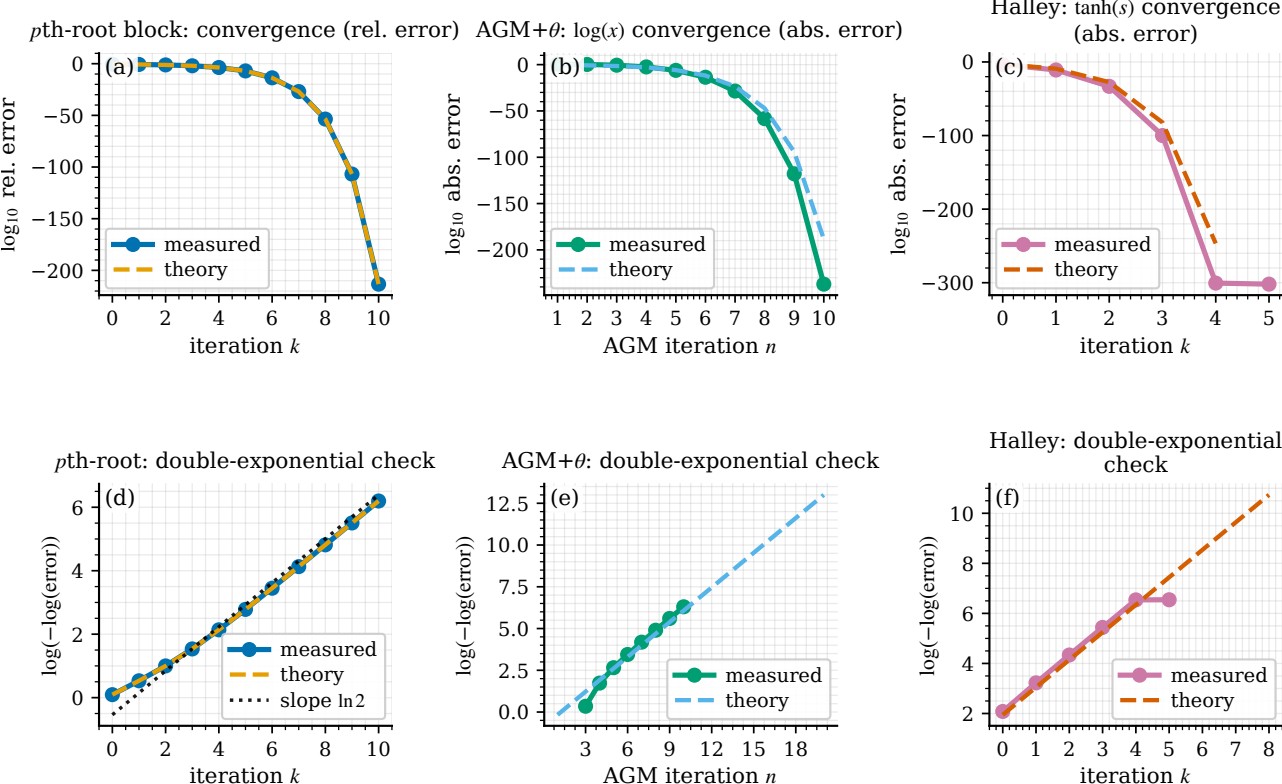

*Figure 5.* Numerical verification of the convergence behavior of the core blocks used in the GELU construction. Panels (a,d) correspond to the composite $p$th-root iteration from Step 1, (b,e) to the AGM-theta logarithm construction from Step 2, and (c,f) to the Halley-based $\tanh$ construction from Step 4. The top row reports $\log_{10}$ error decay together with the corresponding theoretical rate guides, while the bottom row plots $\log(-\log(\text{error}))$, which linearizes doubly exponential convergence and exhibits the expected slopes (approximately $\ln 2$ for the root and AGM-theta blocks, and $\ln 3$ for Halley).

*Consequently, given any $0 < \varepsilon < 1$, it suffices to take $k = \Theta\big(\log\log(1/\varepsilon)\big)$ to obtain*

$$\sup_{x \in [\alpha_0^p, 1]} \big|\tilde{f}_k(x) - x^{1/p}\big| \leq \varepsilon.$$

*Moreover, each update in Equation* (4) *is a fixed bivariate rational map of uniformly bounded degree, so the composite map $x \mapsto \tilde{f}_k(x)$ can be implemented by a constant-width rational neural network whose total size grows like $\Theta(\log\log(1/\varepsilon))$.*

*Proof.* The analysis in Gawlik & Nakatsukasa (2021), applied with the fixed initial endpoint parameter $\alpha_0$, shows that the scaled iterates satisfy the equioscillation identity

$$\max_{x \in [\alpha_0^p, 1]} \frac{\tilde{f}_k(x) - x^{1/p}}{x^{1/p}} = -\min_{x \in [\alpha_0^p, 1]} \frac{\tilde{f}_k(x) - x^{1/p}}{x^{1/p}} = \frac{1 - \alpha_k}{1 + \alpha_k} =: \varepsilon_k, \qquad k \geq 0,$$

where $(\alpha_k)$ is the sequence generated from $\alpha_0$ by Equation (4). Equivalently,

$$\max_{x \in [\alpha_0^p, 1]} \left| \frac{\tilde{f}_k(x) - x^{1/p}}{x^{1/p}} \right| = \varepsilon_k.$$

Moreover, the scalar iteration for $(\alpha_k)$ yields a doubly exponential decay

$$\varepsilon_k = \frac{1 - \alpha_k}{1 + \alpha_k} \leq C_p \exp\big(-c_p\, 2^k\big),$$

for suitable constants $C_p, c_p > 0$ depending only on $p$ and $\alpha_0$; see, e.g., Gawlik & Nakatsukasa (2021, Thm. 3.2). Hence

$$\sup_{x \in [\alpha_0^p, 1]} \left| \frac{\tilde{f}_k(x) - x^{1/p}}{x^{1/p}} \right| \leq C_p \exp\big(-c_p\, 2^k\big),$$

which proves Equation (6).

To achieve a target absolute accuracy $\varepsilon$, note that $0 < x^{1/p} \leq 1$ on $[\alpha_0^p, 1]$. Therefore

$$\sup_{x \in [\alpha_0^p, 1]} \big|\tilde{f}_k(x) - x^{1/p}\big| \leq \sup_{x \in [\alpha_0^p, 1]} \left| \frac{\tilde{f}_k(x) - x^{1/p}}{x^{1/p}} \right|.$$

Thus it is enough to enforce

$$C_p \exp\big(-c_p\, 2^k\big) \leq \varepsilon,$$

which is equivalent to

$$2^k \gtrsim \frac{1}{c_p} \log \frac{C_p}{\varepsilon} \quad \Longrightarrow \quad k \gtrsim \log\log \frac{1}{\varepsilon},$$

so $k = \Theta(\log\log(1/\varepsilon))$ layers suffice.

Finally, for each fixed $k$, the scalars $\alpha_k$ and $\mu(\alpha_k)$ are determined only by $p$ and the initial value $\alpha_0$, and may be precomputed as layer-dependent constants. With these constants fixed, the update

$$(x, f_k) \longmapsto f_{k+1} = \frac{1}{p}\left( (p-1)\,\mu(\alpha_k)\, f_k + \frac{x}{\mu(\alpha_k)^{p-1} f_k^{p-1}} \right)$$

is a rational map whose numerator and denominator degrees are bounded by constants depending only on $p$ and not on $k$ or $\varepsilon$. Thus, keeping $x$ as an input channel and propagating $f_k$ as a state variable, each iteration can be realized by a constant-size rational block with layer-dependent coefficients. Composing $k = \Theta(\log\log(1/\varepsilon))$ such blocks yields a constant-width rational realization of total size $\Theta(\log\log(1/\varepsilon))$ that approximates $x^{1/p}$ to accuracy $\varepsilon$ on $[\alpha_0^p, 1]$. $\square$

We denote by $R_{p\text{-root}}(x)$ the resulting constant-width rational neural network obtained by composing the update Equation (4) a total of $k = \Theta(\log\log(1/\varepsilon))$ times and reading out the scalar $\tilde{f}_k(x)$ in Equation (5). By Theorem A.1, $R_{p\text{-root}}(x)$ satisfies

$$\sup_{x \in [\alpha_0^p, 1]} \big|R_{p\text{-root}}(x) - x^{1/p}\big| \leq \varepsilon,$$

with network size $\Theta(\log\log(1/\varepsilon))$.

STEP 2: AGM-THETA APPROXIMATION OF $\log(x)$ AWAY FROM 1

We now construct a constant-width rational neural network that approximates $\log(x)$ on a fixed domain separated from $x = 1$ by combining the square-root block from Step 1 (the case $p = 2$ of Theorem A.1) with an AGM–theta representation of the logarithm due to Sasaki and Kanada (Sasaki & Kanada, 1982); for the corrected $q^4$-form used below, see Brent 2020, Eq. (40), and for the underlying theta-function and elliptic-integral identities see Borwein & Borwein 1987, Chs. 2 and 7. Fix a separation parameter $0 < \sigma < 1$ and a small positive cutoff $\tau > 0$ and consider

$$\mathcal{D}_{\sigma,\tau} := \{x > 0 : |x - 1| \geq \sigma, \ \tau \leq x \leq 1/\tau\},$$

so that $\mathcal{D}_{\sigma,\tau}$ excludes a small neighbourhood of 1 and extremely small or large values of $x$. Our goal is to construct, for any $0 < \varepsilon < 1$, a constant-width rational neural network $R_{\log}(x)$ that approximates $\log(x)$ to within $\varepsilon$ on $\mathcal{D}_{\sigma,\tau}$ with total size $\Theta((\log\log(1/\varepsilon))^2)$. Notice that for $x \in \mathcal{D}_{\sigma,\tau} \cap (0,1)$ we can use the reciprocity $\log(x) = -\log(1/x)$, so it suffices to construct an approximation for $\log(y)$ when $y \geq 1$ and $|y - 1| \geq \sigma$. We set $q = 1/y$, so $q$ ranges over a compact subinterval of $(0,1)$ determined by $\sigma$ and $\tau$, and $\log(y) = \log(1/q)$.

The construction has two AGM stages:

1. The first evaluates the Jacobi theta functions $\theta_2(0, q^4)$ and $\theta_3(0, q^4)$ via the complete elliptic integral, using standard theta–elliptic-integral identities and the AGM representation of the complete elliptic integral; see Borwein & Borwein (1987, Chs. 2 and 7) and Brent (2020).

   To describe this stage, recall the standard relations between the theta functions, the elliptic modulus $k \in (0,1)$ corresponding to $q$, and the complete elliptic integral $K(k)$; see Borwein & Borwein (1987, Chs. 2 and 7). One has

   $$\theta_3(0,q)^2 = \frac{2}{\pi}K(k), \qquad \theta_2(0,q)^2 = \frac{2}{\pi}k\,K(k),$$

   and

   $$K(k) = \frac{\pi}{2\,\mathrm{AGM}(1, \sqrt{1 - k^2})}.$$

   Thus, given $q$, we first compute the associated modulus $k$, then start with

   $$a_0^{\mathrm{ell}} = 1, \qquad b_0^{\mathrm{ell}} = \sqrt{1 - k^2},$$

   and perform $M_\sigma$ AGM steps

   $$a_{j+1}^{\mathrm{ell}} = \frac{a_j^{\mathrm{ell}} + b_j^{\mathrm{ell}}}{2}, \qquad b_{j+1}^{\mathrm{ell}} = \sqrt{a_j^{\mathrm{ell}} b_j^{\mathrm{ell}}},$$

   to obtain an approximation $K_{M_\sigma}$ to $K(k)$ and hence approximations to $\theta_2(0, q^4)$ and $\theta_3(0, q^4)$ by the formulas above and the standard nome-rescaling (Borwein & Borwein, 1987, Chs. 2 and 7).

2. In the second stage we apply the real AGM directly to the pair

   $$a_0^{\log} = \theta_2(0, q^4)^2, \qquad b_0^{\log} = \theta_3(0, q^4)^2,$$

   where in practice we insert the outputs of the first stage. We then iterate

   $$a_{j+1}^{\log} = \frac{a_j^{\log} + b_j^{\log}}{2}, \qquad b_{j+1}^{\log} = \sqrt{a_j^{\log} b_j^{\log}}, \qquad j \geq 0.$$

   The Sasaki–Kanada identity (Sasaki & Kanada, 1982); see also the corrected $q^4$-form in Brent 2020, Eq. (40)

   $$\log\frac{1}{q} = \frac{\pi/4}{\mathrm{AGM}\big(\theta_2(0,q^4)^2, \theta_3(0,q^4)^2\big)}, \qquad 0 < q < 1 \tag{7}$$

   This is the $q \mapsto q^4$ form of the theta–AGM logarithm formula; the factor $\pi/4$ follows from $\log(1/q^4) = 4\log(1/q)$, together with the symmetry of AGM. It then states that in exact arithmetic

   $$\log\frac{1}{q} = \frac{\pi}{4 \lim_{j\to\infty} a_j^{\log}}.$$

Because $q$ stays in a compact subinterval of $(0, 1)$ when $x \in \mathcal{D}_{\sigma,\tau}$, the arguments of all square roots in both stages lie in a compact interval $[c_{\sigma,\tau}, C_{\sigma,\tau}] \subset (0, \infty)$. To implement this two-stage AGM scheme by a rational neural network we replace each occurrence of $\sqrt{\cdot}$ by the square-root block $R_{2\text{-root}}$ from Theorem A.1. That lemma, applied with $p = 2$ and after rescaling to $[c_{\sigma,\tau}, C_{\sigma,\tau}]$, yields for any $0 < \delta < 1$ a constant-width rational network $R_{2\text{-root}}$ such that

$$\sup_{z \in [c_{\sigma,\tau}, C_{\sigma,\tau}]} \left| R_{2\text{-root}}(z) - \sqrt{z} \right| \leq \delta,$$

with depth $\Theta(\log \log(1/\delta))$ and uniformly bounded degrees in all rational activations.

**Lemma A.2** (AGM-theta approximation of $\log$ away from 1). *Fix a separation parameter $0 < \sigma < 1$ and a (possibly extremely small) positive cutoff $\tau > 0$ and consider $\log(x)$ on $\mathcal{D}_{\sigma,\tau} = \{x > 0 : |x - 1| \geq \sigma, \ \tau \leq x \leq 1/\tau\}$. There exists, for every $0 < \varepsilon < 1$, a constant-width rational neural network $R_{\log}(x)$ such that*

$$\sup_{x \in \mathcal{D}_{\sigma,\tau}} \left| R_{\log}(x) - \log(x) \right| \leq \varepsilon,$$

*and the total network size satisfies*

$$\text{size}(R_{\log}) = \Theta\big((\log \log(1/\varepsilon))^2\big).$$

*Proof.* First consider the idealized scheme in which both AGM stages use exact arithmetic and exact square roots. The classical AGM analysis for complete elliptic integrals shows that there exist constants $C_{\sigma,\tau}^{\text{ell}}, c_{\sigma,\tau}^{\text{ell}} > 0$ such that, after $M_\sigma$ elliptic AGM steps in the first stage, the error in $K(k)$ (and hence in $\theta_2(0, q^4)$ and $\theta_3(0, q^4)$) satisfies

$$\left| K_{M_\sigma}(k) - K(k) \right| \leq C_{\sigma,\tau}^{\text{ell}} \exp\big(-c_{\sigma,\tau}^{\text{ell}} 2^{M_\sigma}\big),$$

see, for example, Borwein & Borwein (1987, Chs. 2 and 7). Likewise, the analysis of the second stage based on Equation (7) yields constants $C_{\sigma,\tau}^{\log}, c_{\sigma,\tau}^{\log} > 0$ such that, after $L$ logarithm AGM steps,

$$\left| \log \frac{1}{q} - \frac{\pi}{4 \, a_L^{\log}} \right| \leq C_{\sigma,\tau}^{\log} \exp\big(-c_{\sigma,\tau}^{\log} 2^L\big),$$

for all $q$ arising from $x \in \mathcal{D}_{\sigma,\tau}$; see Sasaki & Kanada (1982) and Brent (2020, Eq. (40)). Given $0 < \varepsilon < 1$, we may therefore choose

$$M_\sigma = \Theta\big(\log \log(1/\varepsilon)\big), \qquad L = \Theta\big(\log \log(1/\varepsilon)\big),$$

so that the truncation errors from the elliptic and logarithm stages are each at most, say, $\varepsilon/4$ uniformly on $\mathcal{D}_{\sigma,\tau}$. In particular, the total number of AGM steps across both stages is

$$N = M_\sigma + L = \Theta\big(\log \log(1/\varepsilon)\big).$$

We now replace each exact square root by $R_{2\text{-root}}$. In both AGM stages, one iteration sends $(a, b) \in [c_{\sigma,\tau}, C_{\sigma,\tau}]^2$ to

$$\Phi(a, b) = \left( \frac{a + b}{2}, \sqrt{ab} \right),$$

and the implemented map is $\widetilde{\Phi}(a, b) = \left( \frac{a+b}{2}, R_{2\text{-root}}(ab) \right)$, with $\left| R_{2\text{-root}}(z) - \sqrt{z} \right| \leq \delta$ on $[c_{\sigma,\tau}^2, C_{\sigma,\tau}^2]$. Since $\Phi$ is smooth on $[c_{\sigma,\tau}, C_{\sigma,\tau}]^2$ and its derivatives are bounded by constants depending only on $\sigma$ and $\tau$, a standard stability argument shows that the discrepancy between the exact AGM iterates and the inexact ones after $N$ applications of $\widetilde{\Phi}$ is bounded by $C_{\sigma,\tau}' N \delta$ for some $C_{\sigma,\tau}' > 0$ independent of $\varepsilon$. The resulting perturbation in the final value of $\log(1/q)$ is therefore also $O(N\delta)$ uniformly on $\mathcal{D}_{\sigma,\tau}$.

To make this perturbation smaller than, for example, $\varepsilon/2$, it suffices to choose $\delta = \varepsilon/(2C_{\sigma,\tau}' N)$. Since $N = \Theta(\log \log(1/\varepsilon))$ as $\varepsilon \to 0$, this choice implies $\log \log(1/\delta) = \Theta(\log \log(1/\varepsilon))$, because $\delta$ differs from $\varepsilon$ only by a factor that is polynomial in $\log \log(1/\varepsilon)$. Theorem A.1 with $p = 2$ then shows that each square-root block $R_{2\text{-root}}$ achieving accuracy $\delta$ on $[c_{\sigma,\tau}, C_{\sigma,\tau}]$ can be implemented by a constant-width rational network of depth $\Theta(\log \log(1/\delta)) = \Theta(\log \log(1/\varepsilon))$.

Each AGM iteration uses only a constant number of such square-root calls. Therefore one AGM step corresponds to a rational subnetwork of depth $\Theta(\log \log(1/\varepsilon))$. Since the total number of AGM steps across both stages is $N = \Theta(\log \log(1/\varepsilon))$, the overall depth, and hence the total size, of the network implementing $\log(1/q)$ is $\Theta\big((\log \log(1/\varepsilon))^2\big)$, with constant width and uniformly bounded degrees in all rational activations. $\square$

We will use $R_{\log}(x)$ as the logarithm block in the subsequent steps of the GELU construction.

STEP 3: APPROXIMATING artanh ON A SMALL INTERVAL

Using the logarithm block $R_{\log}$ from Theorem A.2, we obtain an artanh block on a small interval around the origin. Fix, for concreteness, $z \in [-\frac{1}{8}, \frac{1}{8}]$ and recall

$$\mathrm{artanh}(z) = \tfrac{1}{2}\big(\log(1+z) - \log(1-z)\big), \qquad |z| < 1.$$

On $[-1/8, 1/8]$ we have $1 \pm z \in [7/8, 9/8]$. For any such $u$ we write $\log(u) = \log(ab) = \log(a) + \log(b)$, choosing a fixed factorization $u = ab$ so that both $a, b \in \mathcal{D}_{\sigma,\tau}$ and each $\log(\cdot)$ on the right is evaluated by $R_{\log}$; if at any stage the argument equals 1 we simply hard-wire $\log(1) = 0$ instead of calling $R_{\log}$. This gives a modified logarithm block $\widetilde{R}_{\log}$ that agrees with $\log$ on $[7/8, 9/8]$ up to error $\varepsilon$. We then define

$$R_{\mathrm{artanh}}(z) := \tfrac{1}{2}\Big(\widetilde{R}_{\log}(1+z) - \widetilde{R}_{\log}(1-z)\Big),$$

which satisfies

$$\sup_{z \in [-1/8, 1/8]} \big|R_{\mathrm{artanh}}(z) - \mathrm{artanh}(z)\big| \leq \varepsilon,$$

and has size $\Theta\big((\log\log(1/\varepsilon))^2\big)$, since we only add $\mathcal{O}(1)$ affine maps and constant corrections on top of $R_{\log}$.

STEP 4: HALLEY ITERATION FOR tanh ON A SMALL INTERVAL

We now construct a constant-width rational neural network that approximates $\tanh$ on a fixed small interval around the origin using the artanh block from Step 3. Fix, for concreteness, an interval $s \in \left[-\frac{1}{8}, \frac{1}{8}\right]$, and define the scalar iteration

$$t_0(s) = s,$$

$$t_{k+1}(s) = t_k(s) - \frac{(1 - t_k(s)^2)\big(\mathrm{artanh}(t_k(s)) - s\big)}{1 - t_k(s)\big(\mathrm{artanh}(t_k(s)) - s\big)}, \qquad k \geq 0. \tag{8}$$

The update Equation (8) is the Halley method applied to the equation $\mathrm{artanh}(t) = s$, written in a form that uses only rational combinations of $t_k(s)$ and $\mathrm{artanh}(t_k(s))$.

**Lemma A.3** (Halley-based approximation of $\tanh$ on a small interval). *Fix $s \in [-1/8, 1/8]$ and let $t_k(s)$ be defined by Equation (8) with exact* artanh*. Then there exist constants $C_{\tanh}, c_{\tanh} > 0$ such that*

$$\sup_{s \in [-1/8, 1/8]} \big|t_k(s) - \tanh(s)\big| \leq C_{\tanh} \exp\big(-c_{\tanh} 2^k\big) \qquad \textit{for all } k \geq 0.$$

*Consequently, given any $0 < \varepsilon < 1$, it suffices to take $k = \Theta\big(\log\log(1/\varepsilon)\big)$ to obtain*

$$\sup_{s \in [-1/8, 1/8]} \big|t_k(s) - \tanh(s)\big| \leq \varepsilon.$$

*Moreover, if each occurrence of* artanh *in Equation (8) is replaced by the block $R_{\mathrm{artanh}}$ from Step 3 (with appropriately chosen accuracy), then $s \mapsto t_k(s)$ can be implemented by a constant-width rational neural network whose total size grows like $\Theta\big((\log\log(1/\varepsilon))^3\big)$.*

*Proof.* For each fixed $s \in [-1/8, 1/8]$, the function $F(t; s) := \mathrm{artanh}(t) - s$ has a unique simple root at $t = \tanh(s)$ with $F'(t; s) = 1/(1 - t^2) \neq 0$ on a neighbourhood of this root. It is classical that Halley's method applied to a scalar equation with a simple root converges locally with at least quadratic (in fact cubic) order; see, e.g., standard analyses of Halley's iteration. Since $s$ ranges over a compact interval and $\tanh(s)$ stays in a compact subinterval of $(-1, 1)$, we may choose a uniform neighbourhood on which the convergence constants can be taken independent of $s$. This yields the doubly exponential error bound

$$\big|t_k(s) - \tanh(s)\big| \leq C_{\tanh} \exp\big(-c_{\tanh} 2^k\big),$$

for suitable $C_{\tanh}, c_{\tanh} > 0$, uniformly in $s \in [-1/8, 1/8]$. The choice $k = \Theta(\log\log(1/\varepsilon))$ then gives the stated $\varepsilon$–accuracy. To obtain a rational neural network, we view Equation (8) as defining a fixed bivariate rational update

$(s, t_k) \longmapsto t_{k+1}$ once $\mathrm{artanh}$ is approximated by the block $R_{\mathrm{artanh}}$. Step 3 shows that for any $0 < \delta < 1$ we can construct $R_{\mathrm{artanh}}$ with

$$\sup_{z \in [-1/8, 1/8]} \left| R_{\mathrm{artanh}}(z) - \mathrm{artanh}(z) \right| \leq \delta,$$

and constant-width network size $\Theta((\log \log(1/\delta))^2)$. Each Halley update then uses a constant number of affine/rational operations and a single call to $R_{\mathrm{artanh}}$, so one iteration corresponds to a constant-width rational subnetwork of size $\Theta((\log \log(1/\delta))^2)$.

As in the stability argument for Theorem A.2, the perturbation introduced by replacing $\mathrm{artanh}$ with $R_{\mathrm{artanh}}$ accumulates at most linearly in the number of iterations: after $k$ steps the discrepancy between the exact and inexact iterates is bounded by $C' k \delta$ for some $C' > 0$ independent of $\varepsilon$. We choose $k = \Theta(\log \log(1/\varepsilon))$, $\delta = \Theta\left(\frac{\varepsilon}{k}\right)$, so that the ideal Halley error and the accumulated approximation error are each at most $\varepsilon/2$ on $[-1/8, 1/8]$. Since $k = \Theta(\log \log(1/\varepsilon))$, this choice ensures $\log \log \frac{1}{\delta} = \Theta(\log \log(1/\varepsilon))$, and therefore each $R_{\mathrm{artanh}}$ call has size $\Theta((\log \log(1/\varepsilon))^2)$. Composing $k = \Theta(\log \log(1/\varepsilon))$ Halley layers gives a constant-width rational neural network of total size

$$\Theta\big(\log \log(1/\varepsilon)\big) \cdot \Theta\big((\log \log(1/\varepsilon))^2\big) = \Theta\big((\log \log(1/\varepsilon))^3\big),$$

with the desired uniform $\varepsilon$–approximation to $\tanh$ on $[-1/8, 1/8]$. $\qquad\square$

We denote by $R_{\tanh}(s)$ the resulting constant-width rational neural network obtained by composing Equation (8) a total of $k = \Theta(\log \log(1/\varepsilon))$ times with $\mathrm{artanh}$ replaced by $R_{\mathrm{artanh}}$, and reading out the scalar $t_k(s)$. By Theorem A.3, $R_{\tanh}(s)$ satisfies

$$\sup_{s \in [-1/8, 1/8]} \left| R_{\tanh}(s) - \tanh(s) \right| \leq \varepsilon,$$

with network size $\Theta\big((\log \log(1/\varepsilon))^3\big)$.

STEP 5: APPROXIMATING THE GELU ACTIVATION

Finally we combine the preceding blocks to approximate the full GELU activation

$$G(x) = \tfrac{x}{2}\big(1 + \tanh(\alpha(x + \beta x^3))\big), \qquad \alpha = \sqrt{2/\pi},\ \beta = 0.044715,$$

on $[-1, 1]$. Set $P(x) = \alpha(x + \beta x^3)$, so $P$ is a fixed polynomial map of constant complexity and $|P(x)| \leq U := \alpha(1 + \beta) < 1$ for $|x| \leq 1$. Choose an integer $m \geq 1$, depending only on $U$, such that $U/2^m \leq 1/8$. Using the double-angle identity $\tanh(x) = 2\tanh(x/2)/(1 + \tanh^2(x/2))$ repeatedly, we can write

$$\tanh(P(x)) = \Psi^{(m)}\Big(\tanh\big(P(x)/2^m\big)\Big), \qquad \Psi(t) = \frac{2t}{1 + t^2},$$

where $\Psi^{(m)}$ is the $m$-fold composition of the fixed rational map $\Psi$. For $|x| \leq 1$ we have $|P(x)/2^m| \leq 1/8$, so $\tanh(P(x)/2^m)$ lies in the domain of the small-interval $\tanh$ block $R_{\tanh}$ from Step 4. Choosing the parameters of $R_{\tanh}$ so that it approximates $\tanh$ on $[-1/8, 1/8]$ to accuracy $\varepsilon'$ and propagating this error through the $m = \mathcal{O}(1)$ applications of $\Psi$ and the outer affine map $x \mapsto \tfrac{x}{2}(1 + \cdot)$ shows that we can achieve

$$\sup_{x \in [-1, 1]} \left| R_{\mathrm{GELU}}(x) - G(x) \right| \leq \varepsilon$$

for some constant-width rational network $R_{\mathrm{GELU}}$, with $\varepsilon'$ chosen as a fixed constant multiple of $\varepsilon$. Since $R_{\tanh}$ has size $\Theta((\log \log(1/\varepsilon))^3)$ by Steps 1–4 and all additional polynomial, affine, and double-angle maps have constant complexity, the overall network $R_{\mathrm{GELU}}$ also has size $\Theta((\log \log(1/\varepsilon))^3)$.

**Fundamental Limits of Rational Approximation for GELU**

We now show that, for any $\varepsilon > 0$, there is no rational neural network of size smaller than $\Omega(\log \log(1/\varepsilon))$ that can approximate the GELU activation function to within $\varepsilon$ on $[-1, 1]$. The argument relies on classical results from complex approximation theory, which relate the best uniform rational error to the distance between the real interval and the complex singularities of the target function.

**Theorem A.4.** *There exists a constant $C > 0$ independent of $n$ such that*

$$\inf_{r_n \in \mathcal{R}_{n,n}} \|\text{GELU} - r_n\|_{L^\infty([-1,1])} \geq C \left( \zeta + \sqrt{1 + \zeta^2} \right)^{-n}, \tag{9}$$

*where $a := \alpha$, $b := \alpha\beta$, and $\zeta > 0$ is the unique positive solution of $a\zeta - b\zeta^3 = -\pi/2$.*

*Proof.* Write $E = [-1, 1]$ and note that the function GELU is meromorphic in $\mathbb{C}$, with poles at the solutions of $az + bz^3 = i(\frac{\pi}{2} + k\pi)$, for $k \in \mathbb{Z}$, where $a := \alpha$ and $b := \alpha\beta$. Let $\zeta > 0$ be the unique positive solution of $a\zeta - b\zeta^3 = -\pi/2$. Then $a(i\zeta) + b(i\zeta)^3 = -i\pi/2$, and set $F_* = \{\pm i\zeta\}$.

By Gonchar (Gonchar, 2011) (see, e.g., §2 and formulas (2.4)–(2.5)), applied to the condenser with plates $E$ and $F_*$, there exists a constant $C > 0$ independent of $n$ such that

$$\inf_{r_n \in \mathcal{R}_{n,n}} \|\text{GELU} - r_n\|_{L^\infty(E)} \geq C \exp\left( -\frac{n}{2} \text{Cap}(E, F_*) \right). \tag{10}$$

Here, $\text{Cap}(E, F_*)$ is the condenser capacity of a condenser with plates $E$ and $F_*$ (Saff & Totik, 1997). There is an explicit expression for $\text{Cap}(E, F_*)$ given by

$$\text{Cap}(E, F_*) = 2\log\left( \zeta + \sqrt{1 + \zeta^2} \right).$$

Substituting this formula into Equation (10) yields Equation (9). $\qquad\square$

Therefore, for any $\varepsilon > 0$ any rational function of degree $n$ with

$$n < \log\left( \frac{C}{\varepsilon} \right) \bigg/ \log\left( \zeta + \sqrt{1 + \zeta^2} \right) = \mathcal{O}\left( \log(1/\varepsilon) \right)$$

cannot approximate GELU uniformly on $[-1, 1]$ to an absolute accuracy of $\varepsilon$. Since a rational neural network with $(3, 2)$ activations of width $\leq W$ and depth $L$ produces a rational function of degree at most $C_{\text{deg}} W 3^L$ for some constant $C_{\text{deg}} > 0$ independent of $W$ and $L$ (see, e.g., (Boullé et al., 2020)), we know that we need

$$C_{\text{deg}} W 3^L \geq \log\left( \frac{C}{\varepsilon} \right) \bigg/ \log\left( \zeta + \sqrt{1 + \zeta^2} \right). \tag{11}$$

The minimum value of $WL$ (the number of parameters in the rational neural network) under the constraint in Equation (11) gives $WL = \Omega\left( \log\log(1/\varepsilon) \right)$.

## B. GELU Neural Network Approximation of Rational Functions

This appendix develops a constructive framework showing that any rational function $R : [-1, 1] \to [-1, 1]$ (with poles away from $[-1, 1]$) can be approximated on $[-1, 1]$ to within any $\varepsilon > 0$ by a GELU network of size $\mathcal{O}(\log^2(1/\varepsilon))$. Moreover, we show that a worst-case family of rational targets cannot be approximated by bounded-parameter GELU networks with size smaller than $\Omega(\log(1/\varepsilon))$.

**GELU network construction**

Our goal in this section is to construct a GELU network that approximates a given rational function $R : [-1, 1] \to [-1, 1]$ with poles outside of $[-1, 1]$. Our argument proceeds in three steps. First, we use analyticity of $R$ on a Bernstein ellipse $E_\rho$ to replace $R$ with its degree-$d$ Chebyshev truncation $p_d$ on $[-1, 1]$, with $d = \Theta(\log(1/\varepsilon))$, so it suffices to approximate $p_d$ to uniform accuracy. Second, we use the fixed GELU map $G$ and a small-argument analysis of $P$ to build a constant-width square block $S_B^{(J)}$ and, via polarization, a product block $\text{Mult}_\delta$, which approximate $u^2$ on $[-B, B]$ and $ab$ on $[-1, 1] \times [-B, B]$ with explicit error bounds and size $\mathcal{O}(\log(1/\delta))$ while keeping all $G$ evaluations in a fixed small-argument regime. Finally, we evaluate $p_d$ by an inexact Clenshaw recurrence using $\text{Mult}_\delta$ for each multiplication $x \cdot (\cdot)$; a stability estimate shows that, for a single choice of $\delta$, the local product errors stay controlled and the resulting constant-width GELU network approximates $R$ on $[-1, 1]$ to accuracy $\varepsilon$ with total size $\mathcal{O}(\log^2(1/\varepsilon))$.

We begin by reducing to a bounded-degree polynomial on $[-1, 1]$. Since $R : [-1, 1] \to [-1, 1]$ is rational with all poles off $[-1, 1]$, there exists a Bernstein ellipse $E_\rho$ with parameter $\rho > 1$ on which $R$ is analytic and $M_\rho := \sup_{z \in E_\rho} |R(z)| < \infty$. Writing the Chebyshev expansion of $R$ and its degree-$d$ truncation as

$$R(x) = \sum_{k=0}^{\infty} c_k T_k(x), \qquad p_d(x) := \sum_{k=0}^{d} c_k T_k(x),$$

standard Bernstein ellipse estimates give (Trefethen, 2019)

$$\|R - p_d\|_{L^\infty([-1,1])} \leq \frac{2M_\rho}{(\rho - 1)} \rho^{-d}, \qquad |c_k| \leq 2M_\rho \rho^{-k}, \quad k \geq 0.$$

Given $0 < \varepsilon < 1$, choose

$$d = \left\lceil \frac{\log\big(8M_\rho/((\rho - 1)\varepsilon)\big)}{\log \rho} \right\rceil,$$

which ensures $\|R - p_d\|_\infty \leq \varepsilon/4$ and $d = \Theta\big(\log(1/\varepsilon)\big)$. In the remaining steps we therefore focus on constructing a constant-width GELU network that evaluates $p_d$ on $[-1, 1]$ with error at most $\varepsilon/4$; combining this with the truncation bound above will yield the desired $\varepsilon$–approximation to $R$.

We now construct the basic arithmetic blocks needed to evaluate $p_d$, which are powers and multiplication, using the fixed one-dimensional GELU activation $G$. The goal of this step is to obtain, for any prescribed range bound $B \geq 1$ and tolerance $\delta \in (0, 1)$, two constant-width GELU subnetworks:

$$S_B^{(J)} : [-B, B] \to \mathbb{R}, \qquad \text{Mult}_\delta : [-1, 1] \times [-B, B] \to \mathbb{R},$$

such that $S_B^{(J)}$ approximates the square map $u \mapsto u^2$ on $[-B, B]$ with relative error $\mathcal{O}(\delta)$, and $\text{Mult}_\delta$ approximates the product map $(a, b) \mapsto ab$ on $[-1, 1] \times [-B, B]$ with absolute error $\mathcal{O}(\delta B^2)$. Both gadgets will be built entirely out of $G$ and affine operations, and every evaluation of $G$ will occur at an input where $P$ stays in a fixed small interval.

We first start with constructing the squaring gadgets, using the identities

$$G(u) - G(-u) = u, \qquad G(u) + G(-u) = u \tanh(P(u)),$$

we introduce a one-parameter family

$$S_s(u) := \frac{G(su) + G(-su)}{\alpha s^2},$$

and analyze $S_s$ for small $s$. A Taylor expansion of $\tanh$ and $P$ in this regime shows that, for $|u| \leq 1$ and $|s|$ sufficiently small,

$$S_s(u) = u^2 + \sum_{\ell \geq 1} h_{\ell-1}(u) s^{2\ell},$$

where the coefficients $h_{\ell-1}$ are analytic in $u$ and uniformly bounded on $|u| \leq 1$. We then apply a finite Richardson expansion in the scaling parameter $s$ to cancel the first $J$ powers of $s^2$ in this expansion, forming

$$T_J(u; s) = \sum_{j=0}^{J} a_j(\gamma) S_{s\gamma^{-j}}(u),$$

with coefficients $a_j(\gamma)$ chosen so that $T_J(u; s) - u^2 = \mathcal{O}(s^{2(J+1)})$ uniformly on $[-1, 1]$. Finally, a rescaling in $u$ yields the square block $S_B^{(J)}(u) = B^2 T_J(u/B; s)$, which approximates $u^2$ on $[-B, B]$ with exponentially small error in $J$.

**Lemma B.1** (GELU-based square block on a fixed range). *For any $B \geq 1$ and any $\delta \in (0, 1)$ there exists a constant-width GELU network $S_B^{(J)} : [-B, B] \to \mathbb{R}$, with $J = \Theta(\log(1/\delta))$, such that*

$$\sup_{u \in [-B,B]} \left| S_B^{(J)}(u) - u^2 \right| \leq \delta B^2.$$

*The realization uses $\mathcal{O}(J)$ evaluations of $G$, and every evaluation $G(z)$ occurs at an input $z$ for which $|P(z)|$ lies in a fixed small interval depending only on $\alpha, \beta$ (and not on $B$ or $\delta$).*

*Proof.* Recall the one-parameter family $S_s(u)$ constructed from $G$ by combining $G(u)$ and $G(-u)$ via the identities for $G(u) \pm G(-u)$. Set $s_0 := \frac{1}{2\alpha(1+|\beta|)}$. Then for $|u| \leq 1$ and $|s| \leq s_0$,

$$|P(su)| = \left| \alpha(su + \beta s^3 u^3) \right| \leq \alpha(1 + |\beta|)|s| \leq \tfrac{1}{2},$$

so $\tanh(P(su))$ is evaluated in a fixed disk where it is analytic. Since $G$ and the construction of $S_s$ depend analytically on $P(su)$ and are even in $s$, the map $s \mapsto S_s(u)$ admits an expansion

$$S_s(u) = u^2 + \sum_{\ell \geq 1} h_{\ell-1}(u) \, s^{2\ell}, \qquad |u| \leq 1, \; |s| \leq s_0,$$

with analytic coefficients $h_{\ell-1}(u)$. By Cauchy estimates there exists $M_H > 0$, depending only on $\alpha, \beta$, such that

$$\sup_{|u| \leq 1} |h_{\ell-1}(u)| \leq M_H \, s_0^{-2\ell} \qquad (\ell \geq 1).$$

Fix $s_* \in (0, s_0]$. For any $0 < s \leq s_*$, the tail is bounded by a geometric series:

$$\sup_{|u| \leq 1} \sum_{\ell \geq J+1} |h_{\ell-1}(u)| \, s^{2\ell} \leq \frac{M_H}{1 - (s/s_0)^2} \left( \frac{s}{s_0} \right)^{2(J+1)} \leq C_{\text{tail}} \left( \frac{s}{s_0} \right)^{2(J+1)},$$

where $C_{\text{tail}} := \frac{M_H}{1 - (s_*/s_0)^2}$. Next we apply Richardson extrapolation in the scale parameter. Fix $\gamma > 1$ and define

$$T_0(u; s) = S_s(u), \qquad T_{m+1}(u; s) = \frac{\gamma^{2(m+1)} T_m(u; s/\gamma) - T_m(u; s)}{\gamma^{2(m+1)} - 1}, \quad m \geq 0.$$

By induction on $m$, $T_m(u; s)$ can be written as a linear combination of $S_{s\gamma^{-j}}(u)$, and in particular

$$T_J(u; s) = \sum_{j=0}^{J} a_j(\gamma) \, S_{s\gamma^{-j}}(u),$$

where the coefficients $a_j(\gamma)$ are independent of $u, s$ and satisfy the moment conditions

$$\sum_{j=0}^{J} a_j(\gamma) \, \gamma^{-2\ell j} = 0 \quad (\ell = 1, \dots, J), \qquad \sum_{j=0}^{J} a_j(\gamma) = 1.$$

Solving this Vandermonde system gives the explicit formula

$$a_j(\gamma) = \prod_{\substack{k=0 \\ k \neq j}}^{J} \frac{\gamma^{-2k}}{\gamma^{-2k} - \gamma^{-2j}} = \prod_{\substack{k=0 \\ k \neq j}}^{J} \frac{1}{1 - \gamma^{-2(j-k)}}.$$

Splitting the product over $k < j$ and $k > j$ and extending to $m \to \infty$ yields the uniform bound

$$|a_j(\gamma)| \leq \prod_{m=1}^{\infty} \frac{1}{(1 - \gamma^{-2m})^2} =: C_\gamma^{\text{unif}} < \infty,$$

independent of $J$ and $j$. Using the expansion of $S_s$ and the cancellations encoded in the moment conditions, we obtain

$$T_J(u;s) - u^2 = \sum_{\ell \geq J+1} h_{\ell-1}(u)\, s^{2\ell} \sum_{j=0}^{J} a_j(\gamma)\, \gamma^{-2\ell j}.$$

For each $\ell \geq 1$ we have

$$\sum_{j=0}^{J} |a_j(\gamma)|\, \gamma^{-2\ell j} \leq C_\gamma^{\mathrm{unif}} \sum_{j=0}^{\infty} \gamma^{-2\ell j} \leq \frac{C_\gamma^{\mathrm{unif}}}{1 - \gamma^{-2}},$$

so for $0 < s \leq s_*$,

$$\sup_{u \in [-1,1]} \left| T_J(u) - u^2 \right| \leq C_1 \left( \frac{s}{s_0} \right)^{2(J+1)}, \qquad C_1 := \frac{C_\gamma^{\mathrm{unif}} C_{\mathrm{tail}}}{1 - \gamma^{-2}}.$$

Now choose $s := s_*/\gamma$. Then $0 < s \leq s_*$ and $s\gamma^{-j} \leq s_*$ for all $j \geq 0$, hence $|P(s\gamma^{-j}u)| \leq \frac{1}{2}$ whenever $|u| \leq 1$. Thus every evaluation of $G$ inside the blocks $S_{s\gamma^{-j}}$ uses an input $z$ with $|P(z)| \leq \frac{1}{2}$, a fixed interval depending only on $\alpha, \beta$. Moreover,

$$\sup_{u \in [-1,1]} \left| T_J(u) - u^2 \right| \leq C_1 \left( \frac{s_*}{s_0} \right)^{2(J+1)} \gamma^{-2(J+1)} \leq C_1\, \gamma^{-2(J+1)}.$$

To extend from $[-1,1]$ to $[-B, B]$, set $S_B^{(J)}(u) := B^2\, T_J(u/B; s)$. Then $u/B \in [-1,1]$ for $u \in [-B, B]$, so

$$\sup_{u \in [-B,B]} \left| S_B^{(J)}(u) - u^2 \right| \leq C_2\, B^2\, \gamma^{-2(J+1)},$$

with $C_2 := C_1$ depending only on $\alpha, \beta, \gamma, s_*$. Choosing $J \geq \frac{1}{2\log\gamma} \left( \log \frac{C_2}{\delta} \right)$ gives

$$\sup_{u \in [-B,B]} \left| S_B^{(J)}(u) - u^2 \right| \leq \delta B^2,$$

and this $J$ satisfies $J = \Theta(\log(1/\delta))$ for fixed $\gamma$.

Finally, the network realization is obtained by combining the $(J+1)$ blocks $S_{s\gamma^{-j}}$ on the same scalar input according to the affine combination defining $T_J$, together with the simple input/output scalings that send $u$ to $u/B$ at the entrance and $T_J(u/B; s)$ to $B^2 T_J(u/B; s)$ at the exit. Each block $S_{s\gamma^{-j}}$ uses a fixed number of evaluations of $G$ (two in our construction, corresponding to $G(u)$ and $G(-u)$), so $T_J$ and hence $S_B^{(J)}$ use $\mathcal{O}(J)$ evaluations of $G$ in total. All remaining operations are affine, and the resulting architecture has width bounded by an absolute constant. This completes the proof. $\qquad\square$

Combining Theorem B.1 with polarization, we obtain both a square and a product gadget: $S_B^{(J)}$ approximates $u \mapsto u^2$ on $[-B, B]$ with error $\delta B^2$ and size $\mathcal{O}(J)$, and for $(a, b) \in [-1,1] \times [-B, B]$ we define $\mathrm{Mult}_\delta(a, b) := \frac{1}{2}\big( S_{B'}^{(J)}(a + b) - S_{B'}^{(J)}(a) - S_{B'}^{(J)}(b) \big)$ with $B' := B + 1$ and $J$ chosen so that $S_{B'}^{(J)}$ has accuracy $\delta B'^2$ on $[-B', B']$. The identity $ab = \frac{1}{2}\big((a + b)^2 - a^2 - b^2\big)$ then shows that $\mathrm{Mult}_\delta$ approximates $ab$ on $[-1,1] \times [-B, B]$ with error $\mathcal{O}(\delta B'^2)$ at the same cost and within the same small-argument regime for $P$ as in the square construction.

STEP 3: CLENSHAW EVALUATION AND GELU NETWORK ASSEMBLY

We now use the Chebyshev approximation from Step 1 and the square/product gadgets from Step 2 to build a one-dimensional GELU network that approximates the target rational map $R$ on $[-1,1]$ with prescribed accuracy. The polynomial $p_d$ constructed in Step 1 will be evaluated via an inexact Clenshaw recurrence in which each multiplication by $x$ is realized by the product gadget $\mathrm{Mult}_\delta$. We first recall the exact Clenshaw scheme for the Chebyshev expansion $p_d(x) = \sum_{j=0}^{d} c_j T_j(x)$. Let $U_k$ denote the Chebyshev polynomials of the second kind. The exact backward Clenshaw recurrence for $p_d$ reads

$$b_{d+1}(x) = b_{d+2}(x) = 0, \qquad b_k(x) = 2x\, b_{k+1}(x) - b_{k+2}(x) + c_k, \quad k = d, \ldots, 1.$$

It returns $p_d(x)$ as

$$p_d(x) = c_0 + x\, b_1(x) - b_2(x).$$

Equivalently, if one also defines

$$b_0(x) = 2x\, b_1(x) - b_2(x) + c_0,$$

then

$$p_d(x) = b_0(x) - x\, b_1(x).$$

One verifies that, for $k = 1, \ldots, d$,

$$b_k(x) = \sum_{j=k}^{d} c_j\, U_{j-k}(x),$$

and therefore

$$|b_k(x)| \le \sum_{j=k}^{d} |c_j|\, (j - k + 1) \le \sum_{m=0}^{\infty} \frac{2M_\rho}{\rho^{m+k}}\, (m+1) = \frac{2M_\rho}{\rho^k} \sum_{m=0}^{\infty} \frac{m+1}{\rho^m} = \frac{2M_\rho}{\rho^k} \cdot \frac{1}{(1 - 1/\rho)^2}.$$

Thus $|b_k(x)| \le C_4 M_\rho$ on $|x| \le 1$, where $C_4 := \frac{2}{(1-1/\rho)^2}$. Pick a range parameter $B \ge 2C_4 M_\rho + 1$, $B' := B + 1$, which depend only on $(M_\rho, \rho)$ and are independent of the accuracy target $\epsilon$.

We next introduce an inexact Clenshaw recurrence in which each multiplication by $x$ is implemented by the product gadget $\mathrm{Mult}_\delta$ from Step 2. For a given tolerance $\delta \in (0,1)$ and range parameter $B$, Step 2 provides a constant-width GELU network $\mathrm{Mult}_\delta : [-1,1] \times [-B, B] \to \mathbb{R}$ such that for $(a, b) \in [-1,1] \times [-B, B]$,

$$|\mathrm{Mult}_\delta(a, b) - ab| \le C_\times\, \delta B'^2,$$

for a constant $C_\times > 0$ depending only on the fixed GELU parameters and the square construction. We run the backward Clenshaw recurrence with the exact affine terms but replace each product $x\, \tilde{b}_{k+1}$ by $\mathrm{Mult}_\delta(x, \tilde{b}_{k+1})$:

$$\tilde{b}_{d+1}(x) = \tilde{b}_{d+2}(x) = 0, \qquad \tilde{b}_k(x) = 2\,\mathrm{Mult}_\delta(x, \tilde{b}_{k+1}(x)) - \tilde{b}_{k+2}(x) + c_k, \quad k = d, \ldots, 1.$$

We return the inexact output

$$y_d(x) := c_0 + \mathrm{Mult}_\delta(x, \tilde{b}_1(x)) - \tilde{b}_2(x).$$

To analyze this scheme, define error terms $e_k(x)$ by

$$\tilde{b}_k(x) = b_k(x) + e_k(x), \qquad e_{d+1}(x) = e_{d+2}(x) = 0.$$

Assume for the moment that $|\tilde{b}_{k+1}(x)| \le B$ for all relevant $k$ and all $x \in [-1,1]$. Introduce the local multiplication errors

$$\Delta_k(x) := \mathrm{Mult}_\delta(x, \tilde{b}_{k+1}(x)) - x\, \tilde{b}_{k+1}(x), \qquad k = d, \ldots, 1,$$

so that $|\Delta_k(x)| \le C_\times\, \delta B'^2$ whenever $|x| \le 1$ and $|\tilde{b}_{k+1}| \le B$. From the recurrence and $|x| \le 1$ we obtain

$$e_k(x) = 2x\, e_{k+1}(x) - e_{k+2}(x) + 2\Delta_k(x), \qquad k = d, \ldots, 1,$$

and hence

$$|e_k(x)| \le 2|e_{k+1}(x)| + |e_{k+2}(x)| + C_\times'\, \delta B'^2, \qquad k = d, \ldots, 1,$$

for some $C_\times' > 0$ depending only on $C_\times$. The associated homogeneous recursion $r_k = 2r_{k+1} + r_{k+2}$ has characteristic equation $r^2 = 2r + 1$ with spectral radius $\lambda := 1 + \sqrt{2}$. Standard comparison with the fundamental solutions gives

$$|e_k(x)| \le C_e^\star\, \lambda^{\max\{d-k,0\}}\, C_\times'\, \delta B'^2, \qquad k = 1, \ldots, d+2, \quad x \in [-1,1],$$

for a constant $C_e^\star > 0$ depending only on the gadget bounds and independent of $d$ and $\epsilon$. The inexact output error can now be written as

$$y_d(x) - p_d(x) = \left(\mathrm{Mult}_\delta(x, \tilde{b}_1(x)) - x\, \tilde{b}_1(x)\right) + xe_1(x) - e_2(x).$$

The first term satisfies $|\mathrm{Mult}_\delta(x, \tilde{b}_1) - x\, \tilde{b}_1| \le C_\times\, \delta B'^2$. Using the bounds on $e_k$ and $|x| \le 1$,

$$\|y_d - p_d\|_\infty \le \left(C_\times + C_e^\star\, \lambda^{d-1}\, C_\times' + C_e^\star\, \lambda^{\max\{d-2,0\}}\, C_\times'\right)\delta B'^2 \le C_R^{(4)}\, \lambda^d\, \delta B'^2,$$

for a constant $C_R^{(4)}$ depending only on $(M_\rho, \rho)$ and the gadget bounds and independent of $d, \epsilon$.

It remains to justify the range invariant $|\tilde{b}_k(x)| \leq B$ on $[-1, 1]$ for the states used by the recurrence. Suppose that $\delta \leq c'_* \lambda^{-d}/B'^2$ for a constant $c'_* > 0$ depending only on the same data. Then a backward induction on $k$ using the inequality for $|e_k|$ shows that $|e_k(x)| \leq B/2$ for all $k = 1, \ldots, d+2$ and all $x \in [-1, 1]$. Indeed, for $k = d+1, d+2$ the bound holds trivially, and the recursion for $|e_k|$ combined with the geometric factor $\lambda^{\max\{d-k,0\}}$ implies that the error remains below $B/2$ provided $c'_*$ is chosen small enough. Consequently, for $k = 1, \ldots, d$,

$$|\tilde{b}_k(x)| = |b_k(x) + e_k(x)| \leq |b_k(x)| + |e_k(x)| \leq C_4 M_\rho + B/2 \leq B$$

by the choice of $B \geq 2C_4 M_\rho + 1$. The terminal states $\tilde{b}_{d+1}$ and $\tilde{b}_{d+2}$ are zero, so they also satisfy the same bound. This establishes the range invariant needed to keep every call to $\mathrm{Mult}_\delta$ within $[-B', B']$ and therefore validates the error bound for $\|y_d - p_d\|_\infty$ derived above.

We now assemble the full GELU network and complete the approximation argument. Fix an accuracy target $\epsilon \in (0, 1)$. Step 1 provides a Chebyshev truncation degree

$$d := \left\lceil \frac{\log\big(8M_\rho/((\rho-1)\epsilon)\big)}{\log \rho} \right\rceil$$

such that $\|R - p_d\|_\infty \leq \epsilon/4$ by the Bernstein–ellipse bound, where $M_\rho$ and $\rho > 1$ are determined by the location of the poles of $R$. Choose $B \geq 2C_4 M_\rho + 1$ and $B' := B + 1$ as above. With these ranges in place, Step 2 guarantees that for any $\delta \in (0, 1)$ there exists a constant-width GELU square block $S_B^{(J)}$ with $J = \lceil c\log(1/\delta) \rceil$ such that

$$\sup_{u \in [-B, B]} \big|S_B^{(J)}(u) - u^2\big| \leq \delta B^2,$$

and such that every input to $G$ in this realization lies in the small-argument regime for $P$. Applying the same construction at range $B'$ yields the square block $S_{B'}^{(J)}$ on $[-B', B']$, from which we obtain the product gadget $\mathrm{Mult}_\delta$ via polarization as in Step 2. This single gadget realizes each occurrence of $x \cdot (\cdot)$ in the Clenshaw recurrence for $p_d$.

Running the inexact Clenshaw recurrence with $\mathrm{Mult}_\delta$ produces the states $\tilde{b}_k$ and the output $y_d$ described above. The range and error bounds for the inexact Clenshaw scheme show that if $\delta \leq c'_* \lambda^{-d}/B'^2$, then $|\tilde{b}_k(x)| \leq B$ for all $k$ and $x \in [-1, 1]$, and

$$\|y_d - p_d\|_\infty \leq C_R^{(4)} \lambda^d \delta B'^2.$$

We now choose $\delta$ to satisfy both this range condition and a target on the evaluation error. Specifically, set

$$\delta = \min\left\{ \frac{\epsilon}{4C_R^{(4)} \lambda^d B'^2}, \; c'_* \lambda^{-d}/B'^2 \right\}.$$

Then $\|y_d - p_d\|_\infty \leq \epsilon/4$ and the bound $|\tilde{b}_k| \leq B$ holds. Combining with $\|R - p_d\|_\infty \leq \epsilon/4$ yields

$$\|R - y_d\|_\infty \leq \|R - p_d\|_\infty + \|p_d - y_d\|_\infty \leq \frac{\epsilon}{4} + \frac{\epsilon}{4} \leq \frac{\epsilon}{2},$$

and in particular $|y_d(x)| \leq 1 + \epsilon/2$ on $[-1, 1]$. Define the normalized network output $f(x) = \frac{y_d(x)}{1+\epsilon/2}$. Then $f : [-1, 1] \to [-1, 1]$ and

$$|R(x) - f(x)| \leq |R(x) - y_d(x)| + \left|1 - \frac{1}{1+\epsilon/2}\right| |y_d(x)| \leq \frac{\epsilon}{2} + \frac{\epsilon/2}{1+\epsilon/2}(1 + \epsilon/2) \leq \epsilon,$$

so $f$ is a GELU network realizable with constant width that $\epsilon$-approximates the rational map $R$ on $[-1, 1]$.

Finally we estimate the network size. Every layer has width bounded by an absolute constant and the input is one-dimensional, so the total number of trainable affine parameters is within a constant factor of the number of non-affine units. The Clenshaw stage uses $\Theta(d)$ multiplications by $x$ ($d$ inside the recurrence plus one in the final combination). Each multiplication is implemented by a product gadget using a square block with $J = \Theta\big(\log(1/\delta)\big)$. With the choice of $\delta$ above,

$$\log \frac{1}{\delta} = \Theta\big(\log(1/\epsilon) + d\big) = \Theta\big(\log(1/\epsilon)\big),$$

since $d = \Theta\big(\log(1/\epsilon)\big)$. Thus each multiplication costs $\mathcal{O}\big(\log(1/\epsilon)\big)$ non-affine units at constant width. Therefore the overall size is

$$\mathcal{O}\big(d \log(1/\delta)\big) = \mathcal{O}\big(\log(1/\epsilon) \log(1/\epsilon)\big) = \mathcal{O}\big(\log^2(1/\epsilon)\big).$$

The realization maintains constant width throughout (the Clenshaw pass carries only $(b_{k+1}, b_{k+2})$ and a constant number of work registers; the $T_J$ sums are computed sequentially with constant registers). All constants $C_1, C_2, C_4, C_\times, C'_\times, C^\star_e, C^{(4)}_R, c'_*$ depend only on the fixed GELU parameters $\alpha, \beta$, on the Bernstein data $(M_\rho, \rho)$, and on the fixed choice of $\gamma$ and $s_*$, and are independent of $\epsilon$. This completes Step 3 and hence the proof that any rational map $R$ analytic on a Bernstein ellipse around $[-1, 1]$ can be approximated to accuracy $\epsilon$ by a constant-width GELU network of size $\mathcal{O}\big(\log^2(1/\epsilon)\big)$.

### B.1. Fundamental limits of GELU approximation of rationals

The preceding section shows that GELU networks of size $\mathcal{O}(\log^2(1/\varepsilon))$ are expressive enough to approximate rational functions whose poles stay away from $[-1, 1]$ to within $\varepsilon > 0$. We now construct a tolerance-dependent rational target family for which no GELU network with uniformly bounded parameters can achieve approximation error $\varepsilon$ with fewer than $\Omega(\log(1/\varepsilon))$ parameters. Although bounded-parameter GELU networks are smooth, their curvatures are uniformly controlled, which limits how sharply they can reproduce nearby-pole rational structure.

**Theorem B.2.** *Let $0 < \varepsilon < \frac{1}{8}$. There exists a rational function $R_\varepsilon : [-1, 1] \to [0, 1]$ such that for any scalar-input, scalar-output GELU network, $F$, with uniformly bounded weights and biases that satisfies $\|F - R_\varepsilon\|_{L^\infty([-1,1])} \leq \varepsilon$, the size of the network is $\Omega\big(\log(1/\varepsilon)\big)$.*

*Proof.* Let $\eta = \sqrt{\varepsilon}$ and consider

$$R_\varepsilon(x) = \frac{\eta^2}{x^2 + \eta^2}.$$

Then $0 < R_\varepsilon(x) \leq 1$ on $[-1, 1]$, $R_\varepsilon \in C^\infty([-1, 1])$, and $R''_\varepsilon(0) = -2/\eta^2 = -2/\varepsilon$. We quantify the curvature required of any smooth approximation to $R_\varepsilon$ at accuracy $\varepsilon$. Suppose $f \in C^2([-1, 1])$ satisfies $\|f - R_\varepsilon\|_{L^\infty([-1,1])} \leq \varepsilon$. Then for any fixed $h \in (0, 1]$, a standard finite-difference form of Taylor's theorem yields a $\xi \in (-h, h)$ such that

$$f(h) + f(-h) - 2f(0) = f''(\xi)\, h^2, \qquad |f''(\xi)| = \frac{|f(h) + f(-h) - 2f(0)|}{h^2}. \tag{12}$$

Since $R_\varepsilon$ is even, $R_\varepsilon(h) = R_\varepsilon(-h)$, and

$$R_\varepsilon(h) + R_\varepsilon(-h) - 2R_\varepsilon(0) = \frac{2\eta^2}{h^2 + \eta^2} - 2 = -\frac{2h^2}{h^2 + \eta^2}.$$

Using $\|f - R_\varepsilon\|_\infty \leq \varepsilon$ and the triangle inequality,

$$|f(h) + f(-h) - 2f(0)| \geq |R_\varepsilon(h) + R_\varepsilon(-h) - 2R_\varepsilon(0)| - 4\varepsilon = \frac{2h^2}{h^2 + \eta^2} - 4\varepsilon.$$

Combining with Equation (13) gives

$$|f''(\xi)| \geq \frac{2}{h^2 + \eta^2} - \frac{4\varepsilon}{h^2}. \tag{13}$$

Select $h = \eta$ (which lies in $(0, 1]$ since $\eta = \sqrt{\varepsilon} < 1$). Then Equation (13) becomes

$$|f''(\xi)| \geq \frac{1}{\eta^2} - \frac{4\varepsilon}{\eta^2} = \frac{1 - 4\varepsilon}{\eta^2}.$$

Because $0 < \varepsilon < \frac{1}{8}$, we have $1 - 4\varepsilon > \frac{1}{2}$, hence

$$|f''(\xi)| \geq \frac{1}{2\eta^2} = \frac{1}{2\varepsilon}.$$

In particular, if $F$ is any GELU network with $\|F - R_\varepsilon\|_\infty \leq \varepsilon$, then $F \in C^\infty$ and the above applies with $f = F$, giving

$$\sup_{x \in [-1,1]} |F''(x)| \geq \frac{1}{2\varepsilon}. \tag{14}$$

Next, we upper bound the curvature available to a bounded-parameter GELU network. Consider a depth-$L$ fully connected scalar-input, scalar-output network with hidden widths $W_1, \ldots, W_L \in \mathbb{N}$:

$$h^{(0)}(x) = x, \quad z^{(\ell)} = A^{(\ell)} h^{(\ell-1)} + b^{(\ell)}, \quad h^{(\ell)} = G(z^{(\ell)}), \quad F(x) = c^\top h^{(L)} + d,$$

where $A^{(\ell)} \in \mathbb{R}^{W_\ell \times W_{\ell-1}}$ with $W_0 = 1$, $b^{(\ell)} \in \mathbb{R}^{W_\ell}$, $c \in \mathbb{R}^{W_L}$, and all entries of $A^{(\ell)}, b^{(\ell)}, c, d$ have magnitude at most $B \geq 1$, where $B$ is independent of $\varepsilon$. Moreover, $G$ is applied entrywise to vector inputs. For

$$G(z) = \tfrac{z}{2}\big(1 + \tanh(\alpha(z + \beta z^3))\big), \qquad \alpha = \sqrt{2/\pi},\ \beta = 0.044715,$$

one has finite constants

$$C_1 := \sup_{z \in \mathbb{R}} |G'(z)| < \infty, \qquad C_2 := \sup_{z \in \mathbb{R}} |G''(z)| < \infty.$$

Set $C := \max\{1,\, C_1,\, \sqrt{C_2}\}$, so that $|G'(z)| \leq C$ and $|G''(z)| \leq C^2$ for all $z$.

With the mixed operator norm, for any matrix $M$ write $\|M\|_{1\to\infty} = \max_i \sum_j |M_{ij}|$; in particular $\|A^{(\ell)}\|_{1\to\infty} \leq B W_{\ell-1}$ and $\|c\|_1 \leq B W_L$ (where $\|c\|_1 = \sum_i |c_i|$). To propagate derivatives through the layers we track the suprema

$$S_\ell = \sup_{x \in [-1,1],\, 1 \leq i \leq W_\ell} \left|\frac{\partial h_i^{(\ell)}}{\partial x}\right|, \qquad T_\ell = \sup_{x \in [-1,1],\, 1 \leq i \leq W_\ell} \left|\frac{\partial^2 h_i^{(\ell)}}{\partial x^2}\right|.$$

By the chain rule, for each $\ell \geq 1$ and each coordinate $i$,

$$\left|\frac{d}{dx} h_i^{(\ell)}(x)\right| = |G'(z_i^{(\ell)}(x))| \left|\sum_j A_{ij}^{(\ell)} \frac{d}{dx} h_j^{(\ell-1)}(x)\right| \leq C \|A^{(\ell)}\|_{1\to\infty} S_{\ell-1},$$

hence

$$S_\ell \leq C \|A^{(\ell)}\|_{1\to\infty} S_{\ell-1} \leq (CBW_{\ell-1}) S_{\ell-1}, \qquad S_0 = 1,\ T_0 = 0,$$

so $S_\ell \leq \prod_{k=1}^\ell (CBW_{k-1})$. For the second derivatives, using $h_i^{(\ell)\prime\prime} = G''(z_i^{(\ell)})\,(z_i^{(\ell)\prime})^2 + G'(z_i^{(\ell)})\, z_i^{(\ell)\prime\prime}$ and $|z_i^{(\ell)\prime}| \leq \|A^{(\ell)}\|_{1\to\infty} S_{\ell-1}$, $|z_i^{(\ell)\prime\prime}| \leq \|A^{(\ell)}\|_{1\to\infty} T_{\ell-1}$, we obtain

$$T_\ell \leq C^2 \|A^{(\ell)}\|_{1\to\infty}^2 S_{\ell-1}^2 + C \|A^{(\ell)}\|_{1\to\infty} T_{\ell-1} \leq (CBW_{\ell-1})^2 S_{\ell-1}^2 + (CBW_{\ell-1}) T_{\ell-1}.$$

Since $CBW_{\ell-1} \geq 1$, an induction gives $T_\ell \leq \ell \left(\prod_{k=1}^\ell CBW_{k-1}\right)^2$. The output linear map contributes $\|c\|_1$, hence

$$\sup_{x \in [-1,1]} |F''(x)| \leq \|c\|_1 T_L \leq BW_L\, L \left(\prod_{k=1}^L CBW_{k-1}\right)^2 = L\,(CB)^{2L}\, B W_L \prod_{k=1}^{L-1} W_k^2.$$

Using $B \leq CB$ and $W_L \leq W_L^2$ (since $C, B, W_L \geq 1$), this relaxes to

$$\sup_{x \in [-1,1]} |F''(x)| \leq L\,(CB)^{2L+1} \prod_{k=1}^L W_k^2. \tag{15}$$

Comparing Equation (14) and Equation (15) gives

$$\frac{1}{2\varepsilon} \leq \sup_{x \in [-1,1]} |F''(x)| \leq L\,(CB)^{2L+1} \prod_{k=1}^L W_k^2. \tag{16}$$

Let $A := \frac{1}{2\varepsilon}$. From Equation (16) we obtain

$$\prod_{k=1}^{L} W_k^2 \geq \frac{A}{L\,(CB)^{2L+1}} \qquad \implies \qquad \prod_{k=1}^{L} W_k \geq \left(\frac{A}{L\,(CB)^{2L+1}}\right)^{1/2}.$$

To convert a product constraint into parameters we pass to the count of hidden-to-hidden weights $S := \sum_{\ell=1}^{L-1} W_\ell W_{\ell+1}$, a subset of the total parameter count, so size $\geq S$. Since $W_\ell \geq 1$,

$$\prod_{\ell=1}^{L-1}(W_\ell W_{\ell+1}) = W_1 W_L \prod_{k=2}^{L-1} W_k^2 \geq \prod_{k=1}^{L} W_k.$$

By AM–GM on $\{W_\ell W_{\ell+1}\}_{\ell=1}^{L-1}$,

$$\left(\frac{S}{L-1}\right)^{L-1} \geq \prod_{\ell=1}^{L-1}(W_\ell W_{\ell+1}) \geq \left(\frac{A}{L\,(CB)^{2L+1}}\right)^{1/2}.$$

Hence, for $L \geq 2$,

$$S \geq (L-1)\exp\left(\frac{\log A - (2L+1)\log(CB) - \log L}{2(L-1)}\right).$$

Writing $k = \log(CB) \geq 0$ and $a = \log A > 0$ (since $\varepsilon < \frac{1}{8}$ implies $A > 4$), and noting $\frac{2L+1}{2(L-1)} \leq \frac{5}{2}$ and $\frac{\log L}{2(L-1)} \leq \frac{1}{2}$ for $L \geq 2$, we get

$$S \geq e^{-(\frac{5}{2}k+\frac{1}{2})}(L-1)\,e^{a/(2(L-1))}.$$

For any $t > 0$, the function $t \mapsto t\,e^{a/(2t)}$ is minimized at $t = a/2$, yielding $t\,e^{a/(2t)} \geq \frac{a}{2}\,e$, so with $t = L-1$, $S \geq c_0 \log A$ for a constant $c_0 > 0$ depending only on $B$ and $G$.

The single-hidden-layer case follows directly from Equation (16): when $L = 1$, Equation (15) gives $A \leq (CB)^3 W_1^2$, and the total size is at least $2W_1$, so size $\geq c_1 A^{1/2} \geq c_2 \log A$ (since $A > 4$ on $0 < \varepsilon < \frac{1}{8}$), for constants $c_1, c_2 > 0$ depending only on $B$ and $G$.

Therefore, in all cases,

$$\text{size} \geq c \log A = \Omega\big(\log(1/\varepsilon)\big),$$

with a constant $c > 0$ depending only on $B$ and $G$. $\qquad\qquad\qquad\qquad\qquad\qquad\qquad\qquad\qquad$ $\square$

## C. Approximation between rational and GELU networks

This section lifts the one-dimensional approximation results to full feedforward architectures on $[-1,1]^d$. We use the scalar GELU-rational constructions as nodewise building blocks and propagate their errors layer by layer, showing that rational networks with Lipschitz scalar activations can be approximated by GELU networks with only a $\log^2$ overhead in size, and that GELU networks can in turn be approximated by rational networks with a $\log\log^3$ overhead, under uniform bounds on weights and biases.

**Theorem C.1.** *Let $0 < \varepsilon < 1$ and let $\|\cdot\|_1$ denote the vector $1$-norm. The following two statements hold:*

*1. Let $R : [-1,1]^d \to [-1,1]$ be a rational network with $M$ layers and at most $k$ nodes per layer, where each node computes $x \mapsto r(a^\top x + b)$ with a rational function $r : [-1,1] \to [-1,1]$ whose Lipschitz constant on $[-1,1]$ is at most $L$, and where $\|a\|_1 + |b| \leq 1$. Then there exists a GELU network $f : [-1,1]^d \to [-1,1]$ of size*

$$\mathcal{O}\big(kM\,\log^2(S_M(L)/\varepsilon)\big), \qquad S_M(L) := \sum_{j=0}^{M-1} L^j,$$

*such that $\max_{x\in[-1,1]^d} |R(x) - f(x)| \leq \varepsilon$.*

*2. Let $f : [-1,1]^d \to [-1,1]$ be a GELU network with $M$ layers and at most $k$ nodes per layer, where each node computes $x \mapsto G(a^\top x + b)$ with*

$$G(z) = \tfrac{z}{2}\big(1 + \tanh(\alpha(z + \beta z^3))\big), \qquad \alpha = \sqrt{2/\pi},\ \beta = 0.044715,$$

*and where $\|a\|_1 + |b| \le 1$. Let $L_G := \sup\{|G'(t)| : |t| \le 1\}$. Then there exists a rational network $R : [-1,1]^d \to [-1,1]$ of size*

$$\mathcal{O}\big(kM \log\log^3(S_M(L_G)/\varepsilon)\big)$$

*such that $\max_{x \in [-1,1]^d} |f(x) - R(x)| \le \varepsilon$. In particular, one may use the numerical bounds*

$$L_G \le 1.083 \text{ on } [-1,1], \qquad L_G \le 1.129 \text{ on } \mathbb{R},$$

*so that $S_M(L_G) \le S_M(1.083)$ on $[-1,1]$.*

*Proof.* All node outputs and pre-activations remain in $[-1,1]$ because $\|a\|_1 + |b| \le 1$ and each activation maps $[-1,1]$ into $[-1,1]$.

1. (Rational $\to$ GELU) Fix a layer index $J$. Let $H$ be the subnetwork of the rational network $R$ up to layer $J$, and let $H_G$ be obtained from $H$ by replacing, node by node, each rational activation $r_{i,j}$ by a GELU subnetwork $f_{i,j}$ with tolerance $\epsilon_j > 0$:

$$\max_{t \in [-1,1]} |r_{i,j}(t) - f_{i,j}(t)| \le \epsilon_j, \qquad 1 \le j \le J, \ 1 \le i \le k_j.$$

Write

$$y_{i,j+1}(x) := r_{i,j+1}\big(a_{i,j+1}^\top H(x) + b_{i,j+1}\big), \qquad \widehat{y}_{i,j+1}(x) := f_{i,j+1}\big(a_{i,j+1}^\top H_G(x) + b_{i,j+1}\big),$$

and define layerwise errors

$$E_{i,j+1} := \max_{x \in [-1,1]^d} |y_{i,j+1}(x) - \widehat{y}_{i,j+1}(x)|, \qquad E_j := \max_{1 \le i \le k_j} E_{i,j}.$$

Add and subtract $r_{i,j+1}\big(a_{i,j+1}^\top H_G(x) + b_{i,j+1}\big)$ to obtain the two-term decomposition

$$E_{i,j+1} \le \max_x |r_{i,j+1}(a^\top H_G(x) + b) - r_{i,j+1}(a^\top H(x) + b)|$$
$$+ \max_x |f_{i,j+1}(a^\top H_G(x) + b) - r_{i,j+1}(a^\top H_G(x) + b)|.$$

For the second term, the argument $a^\top H_G(x) + b$ lies in $[-1,1]$; hence it is bounded by $\epsilon_{j+1}$. For the first term, since $r_{i,j+1}$ is $L$-Lipschitz on $[-1,1]$,

$$\max_x |r_{i,j+1}(a^\top H_G(x) + b) - r_{i,j+1}(a^\top H(x) + b)| \le L \max_x |a^\top (H_G(x) - H(x))| \le L E_j,$$

using $\|a\|_1 \le 1$. Therefore

$$E_{j+1} \le L E_j + \epsilon_{j+1}, \qquad E_0 = 0.$$

Iterating yields

$$E_{J+1} \le \sum_{m=0}^{J} L^m \epsilon_{J+1-m}.$$

With the uniform budget $\epsilon_j := \varepsilon/S_M(L)$, one has $E_{J+1} \le \varepsilon$ for all $J$, hence $\max_{x \in [-1,1]^d} |R(x) - H_G(x)| \le \varepsilon$ at the output layer. There are at most $\sum_{j=1}^{M} k_j \le Mk$ activations, and by the GELU-approximation-of-rational construction with tolerance $\varepsilon/S_M(L)$ each replacement has size $\mathcal{O}\big(\log^2(S_M(L)/\varepsilon)\big)$. The total size is $\mathcal{O}\big(kM \log^2(S_M(L)/\varepsilon)\big)$.

2. (GELU $\to$ rational) Fix a layer index $J$. Let $H$ be the subnetwork of the GELU network $f$ up to layer $J$, and let $H_R$ be obtained from $H$ by replacing each $G$ with a rational network $\widetilde{R}_\delta$ such that

$$\max_{t \in [-1,1]} |G(t) - \widetilde{R}_\delta(t)| \le \delta, \qquad \widetilde{R}_\delta([-1,1]) \subset [-1,1].$$

Write

$$y_{i,j+1}(x) := G\big(a_{i,j+1}^\top H(x) + b_{i,j+1}\big), \qquad \widehat{y}_{i,j+1}(x) := \widetilde{R}_\delta\big(a_{i,j+1}^\top H_R(x) + b_{i,j+1}\big),$$

and define errors as before. Add and subtract $G\big(a_{i,j+1}^\top H_R(x) + b_{i,j+1}\big)$ to obtain

$$E_{i,j+1} \leq \max_x \big|G(a^\top H_R(x) + b) - G(a^\top H(x) + b)\big| + \max_x \big|\widetilde{R}_\delta(a^\top H_R(x) + b) - G(a^\top H_R(x) + b)\big|$$

$$\leq L_G \max_x \big|a^\top (H_R(x) - H(x))\big| + \delta \leq L_G E_j + \delta,$$

using $\|a\|_1 \leq 1$ and that $G$ is $L_G$-Lipschitz on $[-1, 1]$. Hence $E_{j+1} \leq L_G E_j + \delta$ and $E_M \leq S_M(L_G)\,\delta$. Choosing $\delta := \varepsilon/S_M(L_G)$ yields $\max_{x \in [-1,1]^d} |f(x) - H_R(x)| \leq \varepsilon$. By the rational-approximation-of-GELU construction with tolerance $\delta$, each replacement has size $\mathcal{O}\big(\log \log^3(S_M(L_G)/\varepsilon)\big)$; with at most $Mk$ activations the total size is $\mathcal{O}\big(kM \log \log^3(S_M(L_G)/\varepsilon)\big)$. On $[-1, 1]$ one may take $L_G \leq 1.083$ (and globally $L_G \leq 1.129$), whence $S_M(L_G) \leq S_M(1.083)$ on $[-1, 1]$. $\qquad\square$

## D. Normalization-induced conditioning issues for adaptive rational activations

We expand the discussion from Section 4.5 and record the two mechanisms in a form suitable for analysis. Let $u \in \mathbb{R}^d$ be a pre-activation vector. A normalization layer outputs

$$z \;=\; \gamma \odot \tfrac{u-\mu}{\sigma} \;+\; \beta, \tag{17}$$

where $\mu, \sigma$ are statistics over an index set and $\gamma, \beta$ are learned (implementations typically use $\sigma = \sqrt{\mathrm{Var} + \varepsilon_{\mathrm{eps}}}$). Batch normalization (Ioffe & Szegedy, 2015), layer normalization (Ba et al., 2016), and group normalization (Wu & He, 2018) fit Equation (17). Recall that instead of taking exact rational activation functions (see (Boullé et al., 2020)), we take a "safe" version that excludes real-axis poles by construction, i.e.,

$$r_\theta(z) \;=\; \frac{P_\theta(z)}{Q_\theta(z)}, \qquad P_\theta(z) \;=\; \sum_{j=0}^{m} a_j z^j, \qquad \widetilde{Q}_\theta(z) \;=\; \sum_{k=0}^{n} b_k z^k, \qquad Q_\theta(z) \;=\; 1 + \big|\widetilde{Q}_\theta(z)\big|. \tag{18}$$

Activation functions of this form can absorb affine reparameterizations exactly. For $s \neq 0$ and $t$, let $A(x) = sx + t$. Then there exists $\theta' = \theta'(s, t, \theta)$ such that

$$r_\theta(A(x)) \;=\; r_{\theta'}(x), \tag{19}$$

since $P_\theta(sx + t)$ and $\widetilde{Q}_\theta(sx + t)$ are again polynomials in $x$ of degrees at most $m$ and $n$, and the outer map $y \mapsto 1 + |y|$ preserves the "safe" form in Equation (18).

A concrete invariance direction is visible under scaling. In the scalar case with $\beta = 0$, define

$$\gamma_\eta = e^\eta \gamma, \qquad a_{j,\eta} = e^{-j\eta} a_j, \qquad b_{k,\eta} \;= e^{-k\eta} b_k. \tag{20}$$

Then, by the monomial rescaling identities $P_{\theta_\eta}(e^\eta x) = P_\theta(x)$ and $\widetilde{Q}_{\theta_\eta}(e^\eta x) = \widetilde{Q}_\theta(x)$, we have

$$r_{\theta_\eta}\Big(\gamma_\eta \tfrac{u-\mu}{\sigma}\Big) = r_\theta\Big(\gamma \tfrac{u-\mu}{\sigma}\Big). \tag{21}$$

In the absence of explicit regularization or constraints on $\gamma$ or $\theta$, this produces a flat direction in the joint parameterization $\{\gamma, \beta, \theta\}$, worsening conditioning for first-order training.

Second, for batch normalization during training, $\mu, \sigma$ depend on the mini-batch (Ioffe & Szegedy, 2015), so $z$ is random even for fixed $u$, and this randomness is amplified by polynomial growth and by the coefficient dependence of the safe denominator. The coefficient sensitivities satisfy (for $Q_\theta(z) \neq 0$, i.e., a.e. in $z$)

$$\partial_{a_j} r_\theta(z) \;=\; \tfrac{z^j}{Q_\theta(z)}, \qquad \partial_{b_k} r_\theta(z) \;=\; - \tfrac{P_\theta(z)\, z^k}{Q_\theta(z)^2}\, \mathrm{sgn}\big(\widetilde{Q}_\theta(z)\big), \tag{22}$$

with $\mathrm{sgn}(0)$ understood in the subgradient sense. Hence variance control involves moments such as $\mathbb{E}\big[z^{2j}/Q_\theta(z)^2\big]$ and $\mathbb{E}\big[P_\theta(z)^2 z^{2k}/Q_\theta(z)^4\big]$, together with sensitivity to sign flips of $\mathrm{sgn}(\widetilde{Q}_\theta(z))$ under statistic-dependent perturbations.

In particular, since $Q_\theta(z) \geq 1$,

$$\big|\partial_{a_j} r_\theta(z)\big| \;\leq\; |z|^j, \qquad \big|\partial_{b_k} r_\theta(z)\big| \;\leq\; |P_\theta(z)|\, |z|^k. \tag{23}$$

Thus, while true poles are excluded by construction, gradient-noise amplification is controlled by polynomial moments of the normalized activations and of $P_\theta(z)$, and can be exacerbated by statistic-dependent sign changes induced by the learned denominator map. This mechanism is structurally absent for fixed activations, which have neither trainable coefficients with polynomial sensitivities nor statistic-dependent sign changes tied to a learned denominator.

These effects motivate reporting both normalized and non-normalized settings when comparing adaptive rationals to fixed activations, and suggest that stabilizing variants which reduce non-identifiability or control polynomial moment growth on the observed input range can materially change training behavior.

*Table 6.* Training and evaluation details for the CIFAR-10 VGG study.

| Setting | Baseline | Augmented |
|---|---|---|
| Epochs | 60 | 60 |
| Batch size | 128 | 128 |
| Optimizer | SGD | SGD |
| Base learning rate | 0.02 | 0.02 |
| Momentum | 0.9 | 0.9 |
| Nesterov | True | True |
| Learning-rate schedule | None | Step at epochs 30 and 45, $\gamma = 0.1$ |
| Augmentation | None | Crop plus flip |
| Label smoothing | 0 | 0.1 |
| Mixup $\alpha$ | 0 | 0.2 |
| VGG dropout before classifier | 0 | 0.3 |
| Weight decay on Conv and Linear weights | 0 | $5 \times 10^{-4}$ |
| Weight decay on biases and norm parameters | 0 | 0 |
| Rational coefficient weight decay | 0 | 0 |
| Rational coefficient learning rate | Base learning rate | Base learning rate times multiplier |
| Rational coefficient multiplier | 1 | 0.5 |
| GroupNorm in VGG | Off | On or off |
| GroupNorm placement | None | After Conv2d and before activation |
| Group count | None | 16, adjusted to divide channels |
| Initialization sweep | Not used | Used for every activation, includes LSUV toggle |
| Seeds | 0–4 | 0–4 |
| Selection | Best test accuracy over training | Best test accuracy over training |

# E. CIFAR-10 experiments

CIFAR-10 (Krizhevsky & Hinton, 2009) is a ten-class image classification benchmark of $32 \times 32$ natural images. Inputs are normalized per channel with mean 0.4914, 0.4822, 0.4465 and standard deviation 0.2023, 0.1994, 0.2010, using the torchvision CIFAR-10 interface. We run two pipelines. The baseline pipeline uses no data augmentation, no label smoothing, no Mixup, no dropout, no normalization layers in VGG, no weight decay, and a fixed learning rate. The augmented pipeline adds random crop with padding 4 and horizontal flip, label smoothing, Mixup, dropout, and a weights-only weight-decay policy that decays convolution and linear weights while setting weight decay to zero for biases and normalization parameters. In the augmented pipeline, GroupNorm is toggled on or off and, when enabled, is applied in VGG with a target of 16 groups reduced when needed to divide channel count.

We evaluate ReLU, LeakyReLU, GELU, Swish as SiLU, and Rational from `rational-activations` (Delfosse et al., 2020). For the baseline results, Rational uses a LeakyReLU target initialization. For the augmented results, we report two rational target initializations, LeakyReLU and GELU, denoted Rational 1 and Rational 2. Model initialization is treated as a controlled factor in the augmented pipeline with exactly two settings: the default initialization and the optional LSUV initialization toggle. When LSUV is disabled, we use the standard PyTorch default initialization for the model parameters.

Under the experimental protocol and hyperparameter settings summarized above, for a finer-grained view beyond the final best accuracies in Tables 1 and 2, Figure 6 plots the epoch-wise top-1 test accuracy curves for the same six settings (plain; boosted without GroupNorm; boosted with GroupNorm) for both VGG4 and VGG8, showing mean across five seeds with one standard deviation shading.

Complementing the epoch-wise accuracy curves in Figure 6, we also visualize how the learned Rational nonlinearities evolve across depth during training. For each Rational layer in the VGG-8 feature stack under the augmented training recipe without GroupNorm, we snapshot the scalar activation function at epochs 0, 5, 30, and 60, and overlay the empirical distribution of that layer's pre-activation inputs (estimated from held-out mini-batches). This view separates changes in the learned functional form from changes in the input statistics seen by each layer, and provides a layerwise diagnostic of how the model allocates nonlinearity throughout training.

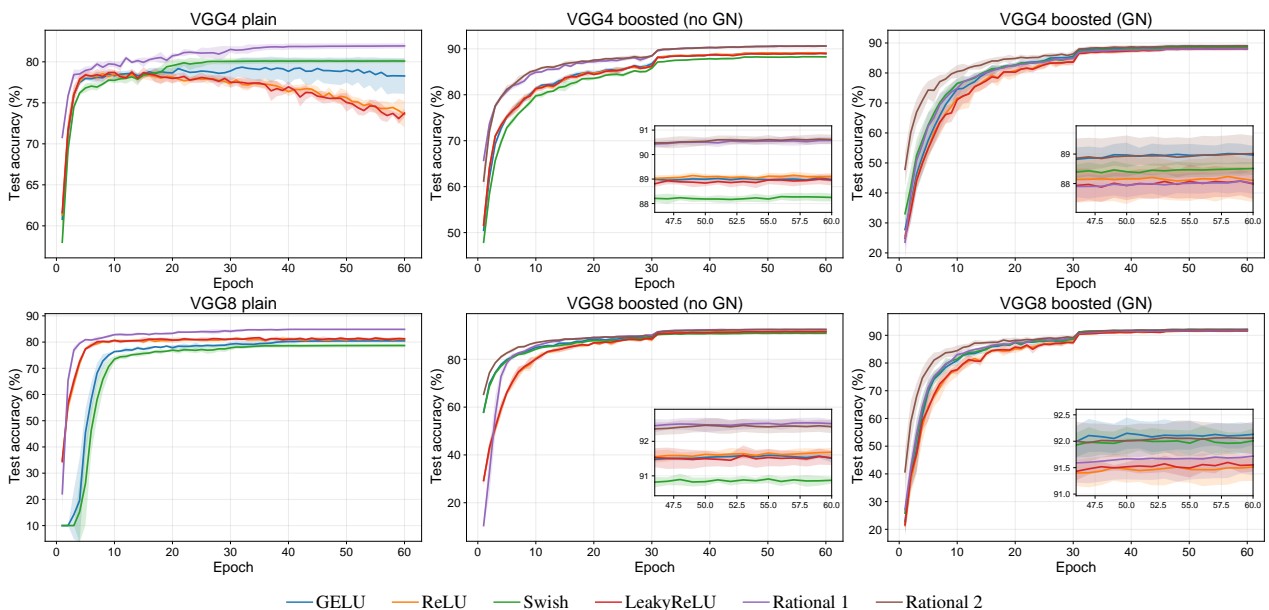

*Figure 6.* CIFAR-10 score curves. Top-1 test accuracy versus epoch. Curves show mean across five seeds and the shaded region is one standard deviation. Columns are baseline, augmented without GroupNorm, and augmented with GroupNorm. Rows are VGG4 (top) and VGG8 (bottom).

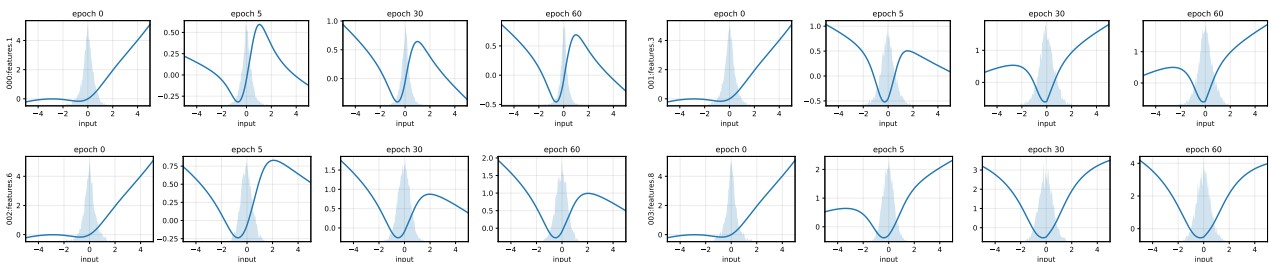

*Figure 7.* Layerwise Rational activation snapshots for VGG8 (pages 1–4). Each panel shows the learned Rational nonlinearity at epochs 0, 5, 30, and 60. The light shaded overlay is the empirical histogram (density) of the corresponding layer input, estimated from held-out batches and plotted on a secondary axis.

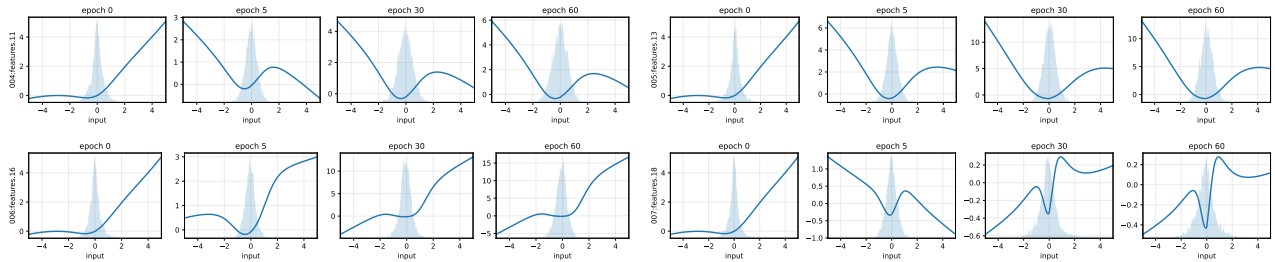

*Figure 8.* Layerwise Rational activation snapshots for VGG8 (pages 5–8). Each panel shows the learned Rational nonlinearity at epochs 0, 5, 30, and 60. The light shaded overlay is the empirical histogram (density) of the corresponding layer input, estimated from held-out batches and plotted on a secondary axis.

## F. Offline reinforcement learning experiments

We study offline reinforcement learning on continuous-control locomotion benchmarks in MuJoCo (Todorov et al., 2012), where the goal is to learn a state-feedback controller for a simulated robot from a fixed dataset of experience without any additional interaction during training (Fu et al., 2021). Each task is a finite-horizon Markov decision process with continuous state $s_t$ (robot configuration and velocities) and continuous action $a_t$ (joint torques), and an episode return $R = \sum_{t=0}^{T-1} r_t$

*Table 7.* Tasks, offline datasets, and evaluation environments used in the offline RL study (Todorov et al., 2012; Younis et al., 2024; Towers et al., 2024).

| Task | Minari dataset | Gymnasium env |
|---|---|---|
| HalfCheetah-medium | `mujoco/halfcheetah/medium-v0` | `HalfCheetah-v5` |
| Hopper-medium | `mujoco/hopper/medium-v0` | `Hopper-v5` |
| Walker2d-medium | `mujoco/walker2d/medium-v0` | `Walker2d-v5` |

over at most $T = 1000$ steps. HalfCheetah requires learning a planar bipedal gait to run forward, Hopper requires balancing and hopping forward on a single leg, and Walker2d requires stabilizing and walking forward with a two-legged agent (Todorov et al., 2012). Rewards are the environment-provided locomotion objectives (e.g., forward-progress terms with control and stability penalties) and are accumulated over the episode to produce the reported return. We train from the Minari medium datasets (Younis et al., 2024), which provide tuples $(s_t, a_t, r_t, s_{t+1}, d_t)$ collected by a partially competent behavior policy, so the learner must improve beyond the behavior policy while remaining robust to distribution shift induced by the fixed data (Fu et al., 2021). Policies are evaluated by executing rollouts in the corresponding Gymnasium v5 environments (Towers et al., 2024), using deterministic action selection for both algorithms. In addition to reporting raw episode return, we report a v5-normalized score to compare performance across tasks on a shared scale. The v5-normalized score uses task-specific random and expert anchors measured under the same Gymnasium v5 evaluation environment and the corresponding Minari expert dataset, respectively, yielding a D4RL-style normalized metric that is consistent with the evaluation API used here (Fu et al., 2021). Concretely, for an average evaluation return $R$, we report

$$\text{Score} = 100 \cdot \frac{R - R_{\text{random}}}{R_{\text{expert}} - R_{\text{random}}}.$$

The expert anchor $R_{\text{expert}}$ is computed as the mean episode return of the corresponding Minari expert dataset for each task, and the random anchor $R_{\text{random}}$ is estimated by rolling out 100 episodes of a random policy in the same Gymnasium v5 evaluation environment. These anchors are cached and reused for subsequent runs under the same environment specification. We include reproducibility details for the offline RL results. All runs use the same pipeline across activations, and we report learning curves as a function of gradient updates. Curves aggregate five seeds with mean and one standard deviation.

We normalize observations using dataset mean and standard deviation computed from the offline dataset and apply the same normalization at evaluation time. We construct transitions $(s, a, r, s', d)$ using the dataset termination flags as $d$ and ignore time-limit truncations when forming terminal transitions. We set `max_action` $= \max_i |(\text{high})_i|$ from the evaluation environment action space, scale dataset actions into $[-1, 1]$ by dividing by `max_action` during training, and rescale policy outputs by `max_action` at evaluation time. We train for $300\,000$ gradient updates with batch size 256 and evaluate every $10\,000$ updates for 20 episodes with a maximum episode length of 1000. Each configuration is run with five seeds.

We evaluate two offline RL algorithms: Implicit Q-Learning (IQL) (Kostrikov et al., 2021) and TD3+BC (Fujimoto & Gu, 2021). For IQL we use a squashed Gaussian actor with a learned, state-independent log standard deviation vector; the actor learning rate follows a cosine schedule over the training horizon, and evaluation uses the mean action. For TD3+BC we use a deterministic actor with delayed policy updates and target-policy smoothing noise as in TD3, together with the behavior-cloning regularizer in TD3+BC (Fujimoto & Gu, 2021). All networks are MLPs with two hidden layers of width 256 and use the same activation throughout the MLP. We sweep activations over ReLU, SiLU, GELU, and Rational; Rational is instantiated from the `rational-activations` package (Delfosse et al., 2020) and uses the initialization target specified in the provided code, with the rational-coefficient learning-rate multiplier set to one. For IQL only, we scale dataset rewards by $1/(R_{\text{max}} - R_{\text{min}})$, where $R_{\text{max}}$ and $R_{\text{min}}$ are the maximum and minimum episode returns in the offline dataset; TD3+BC uses unscaled rewards.

*Table 8.* Hyperparameters for the RL study. Values are shared across tasks unless noted (Kostrikov et al., 2021; Fujimoto & Gu, 2021).

| Setting | Value |
| --- | --- |
| Updates | 300 000 |
| Batch size | 256 |
| Network widths | 256–256 |
| Optimizer | Adam |
| Learning rate | $3 \times 10^{-4}$ |
| Discount $\gamma$ | 0.99 |
| Target update rate | 0.005 |
| Evaluation interval | 10 000 updates |
| Evaluation episodes | 20 |
| Max episode length | 1000 |
| Seeds | 0–4 |
| Rational LR multiplier | 1 |
| IQL expectile | 0.7 |
| IQL inverse temperature | 3.0 |
| IQL advantage clip | 100 |
| IQL actor schedule | cosine over 300 000 updates |
| TD3+BC policy noise | 0.2 |
| TD3+BC noise clip | 0.5 |
| TD3+BC policy frequency | every 2 critic updates |
| TD3+BC BC weight $\alpha$ | 2.5 |

Under the experimental protocol and hyperparameter settings summarized above, Figure 9 plots learning curves of evaluation performance as a function of gradient updates. Each panel shows the v5-normalized score computed from the evaluation return using the task-specific anchors described above, with curves reporting the mean across five seeds and one standard deviation shading. We evaluate every 10 000 updates for 20 episodes in the Gymnasium v5 environment and plot the resulting scores against training progress, enabling a comparison of both convergence speed and final performance.

Complementing the learning curves in Figure 9, we also visualize how the learned Rational nonlinearities evolve during training for IQL on HalfCheetah-medium with Rational initialized to ReLU. For each Rational layer in the IQL networks, including the two hidden-layer nonlinearities in each critic $Q_1$ and $Q_2$, the value network $V$, and the actor mean network, we snapshot the scalar activation function at updates 50k, 100k, 150k, and 300k, and overlay the empirical distribution of that layer's pre-activation inputs estimated from held-out mini-batches of offline transitions. This view separates changes in the learned functional form from changes in the input statistics induced by the evolving networks, and provides a module-wise diagnostic of how IQL allocates nonlinearity under offline data constraints.

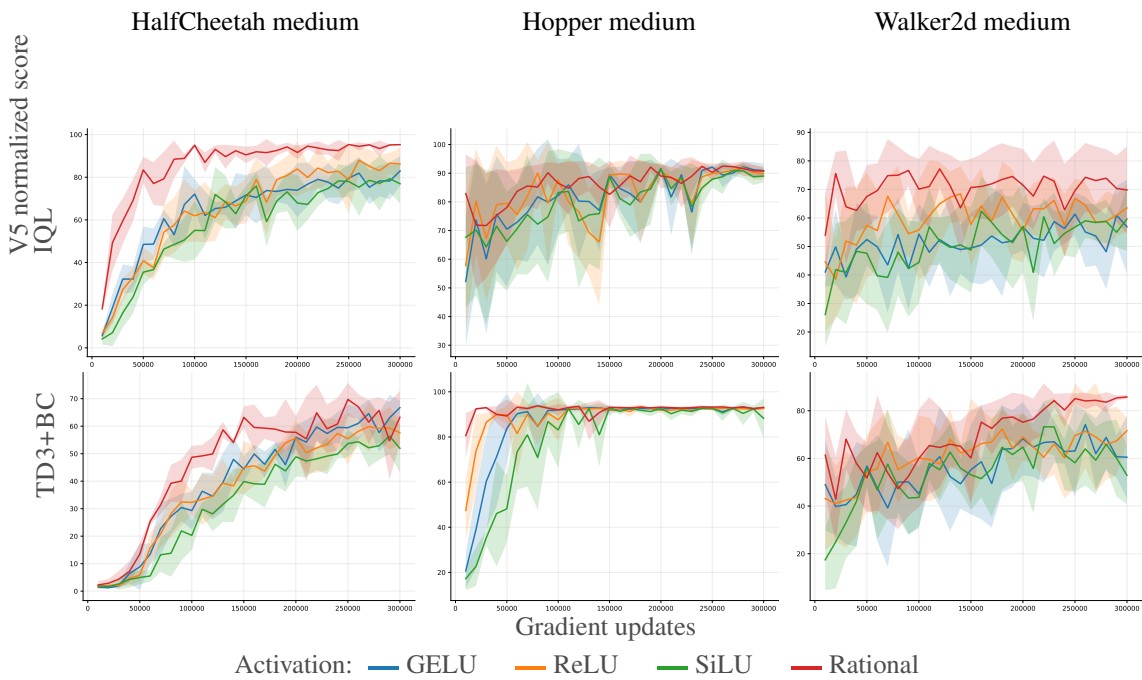

*Figure 9.* Offline reinforcement learning curves on medium locomotion benchmarks (Todorov et al., 2012; Kostrikov et al., 2021; Fujimoto & Gu, 2021). Each panel plots the v5-normalized evaluation score versus gradient updates. For each activation, the solid curve is the mean over five random seeds and the shaded band corresponds to one standard deviation. Columns correspond to environments (HalfCheetah medium, Hopper medium, Walker2d medium). Rows correspond to algorithms (IQL top row and TD3+BC bottom row).

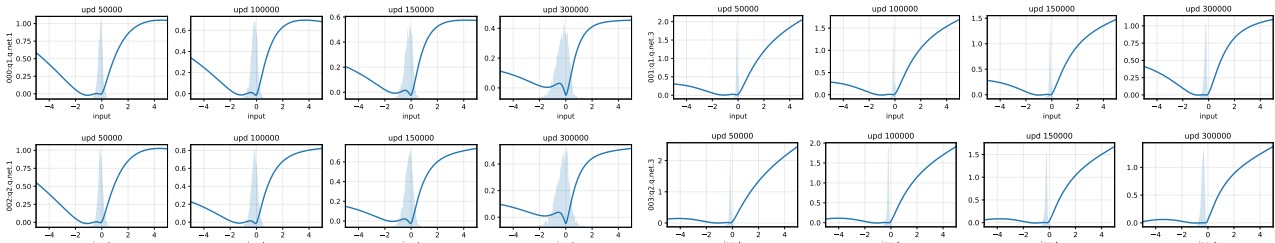

*Figure 10.* Rational activation snapshots for IQL on HalfCheetah-medium with Rational initialized to ReLU, critic networks. Each panel shows the learned Rational nonlinearity at updates 50k, 100k, 150k, and 300k. The light shaded overlay is the empirical histogram (density) of the corresponding layer pre-activation input, estimated from held-out mini-batches of offline transitions. From top-left to bottom-right: $Q_1$ hidden layer 1, $Q_1$ hidden layer 2, $Q_2$ hidden layer 1, and $Q_2$ hidden layer 2.

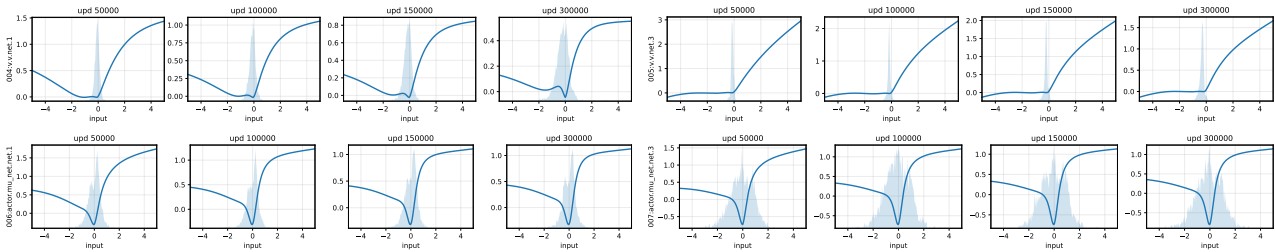

*Figure 11.* Rational activation snapshots for IQL on HalfCheetah-medium with Rational initialized to ReLU, value and actor networks. Each panel shows the learned Rational nonlinearity at updates 50k, 100k, 150k, and 300k. The light shaded overlay is the empirical histogram (density) of the corresponding layer pre-activation input, estimated from held-out mini-batches of offline transitions. From top-left to bottom-right: $V$ hidden layer 1, $V$ hidden layer 2, actor mean network hidden layer 1, actor mean network hidden layer 2.

## G. Tiny ImageNet transformer experiments

We study rational activations for image classification on Tiny ImageNet, a 200 class dataset with 100000 training images and 10000 validation images. We train from scratch on 64 by 64 inputs and report top 1 validation accuracy using an exponential moving average of model weights. We take the best EMA validation accuracy over the full training run and report total wall clock training time in minutes.

We evaluate three transformer families implemented in `timm`: ViT Small with patch size 8 (Dosovitskiy et al., 2021), CaiT Small 24 (Touvron et al., 2021), and Swin Tiny (Liu et al., 2021). For each family, we keep the model definition and optimizer fixed and change only the element wise activation in the MLP blocks. We include reproducibility details for the Tiny ImageNet transformer results. All runs use the same training pipeline across activations and differ only in the MLP nonlinearity and, for the normalization free ViT setting, the presence or absence of ViT block normalization.

Images are converted to RGB and normalized using ImageNet mean and standard deviation. For training augmentation, we use a mild random resized crop tuned for 64 by 64 resolution, random horizontal flips, color jitter, RandAugment, and random erasing. We additionally use Mixup and CutMix with label smoothing, following a DeiT style recipe adapted to this resolution. We use repeated augmentation with three repeats per sample per epoch implemented through a sampler that repeats indices and relies on fresh stochastic augmentation per access.

We train for 100 epochs with AdamW and a cosine learning rate schedule with linear warmup over the first 5 epochs. We clip gradients to norm 1.0 and use drop path rate 0.1. We evaluate once per epoch on the validation split and track EMA weights with decay 0.999, reporting the EMA top 1 accuracy.

We sweep ReLU, GELU, and a trainable low degree rational activation. The rational nonlinearity is instantiated from the `rational-activations` package (Delfosse et al., 2020) with degrees 5 and 4 and version A, initialized to match GELU. Rational coefficients are placed in a separate AdamW parameter group with the model-specific learning-rate multipliers in Table 10 and weight decay zero. We additionally test a normalization free ViT S 8 variant motivated by recent evidence that transformer normalization can be removed when paired with an appropriate adaptive element wise nonlinearity (Zhu et al., 2025). In this modified model, we bypass the ViT block normalization modules `block.norm1` and `block.norm2` in every transformer block for the full training run, while keeping the data pipeline, optimizer, and schedule unchanged. We apply the same normalization bypass mechanism across the activation sweep. Under this unchanged no-norm setup, the ReLU and GELU runs diverged to NaNs early in training and produced no usable checkpoint, while the rational run remained stable; the main results table therefore reports these fixed-activation runs as NaN.

*Table 9.* Models, data source, and evaluation split used in the Tiny ImageNet transformer study. All runs use `timm` with `pretrained` set to false, `num_classes` set to 200, and `img_size` set to 64.

| Model family | `timm` identifier | Data split |
|---|---|---|
| ViT S 8 | `vit_small_patch8_224` | Tiny ImageNet val |
| CaiT S24 | `cait_s24_224` | Tiny ImageNet val |
| Swin T | `swin_tiny_patch4_window7_224` | Tiny ImageNet val |

*Table 10.* Hyperparameters for the Tiny ImageNet transformer study. Values are shared across model families unless noted.

| Setting | Value |
|---|---|
| Epochs | 100 |
| Batch size | 128 |
| Optimizer | AdamW |
| Base learning rate | $2 \times 10^{-3}$ scaled by effective batch over 512 |
| Minimum learning rate | $1 \times 10^{-6}$ |
| Warmup | 5 epochs |
| Weight decay | 0.05 |
| Gradient clipping | 1.0 |
| Drop path rate | 0.1 |
| Label smoothing | 0.1 |
| Mixup alpha | 0.8 |
| CutMix alpha | 1.0 |
| RandAugment | 2 ops, magnitude 9 |
| Random erasing probability | 0.25 |
| Repeated augmentation repeats | 3 |
| EMA decay | 0.999 |
| Activation sweep | ReLU, GELU, Rational |
| Rational degrees | 5 and 4 |
| Rational version | A |
| Rational initialization target | GELU |
| Rational coefficient learning rate multiplier | 2 for CaiT, 4 for ViT, and 8 for Swin |
| Rational coefficient weight decay | 0 |

