# OpenReview forum: "Rational Neural Networks have Expressivity Advantages"
_ICML.cc/2026/Conference — ICML 2026 regular_

### Official Review · Reviewer_pPPX · 2026-02-18

**Soundness:** 3
**Presentation:** 3
**Significance:** 2
**Originality:** 2
**Overall Recommendation:** 4
**Confidence:** 4

**Summary:**

This paper studies neural networks with trainable rational activation functions i.e., those which use $p(x)/q(x)$ where $p$ and $q$ are polynomials of fixed degrees.  In the theoretical analysis, they compare the expressivity of these networks with the non-rational counterparts e.g., GELU. They also provide several experiments to confirm their analysis.

**Compliance With Llm Reviewing Policy:**

Affirmed.

**Final Justification:**

The authors addressed most of my concerns and questions in the rebuttal, and I have therefore increased my score. Incorporating the related works and adding arguments regarding sample complexity would make the paper more complete.

**Key Questions For Authors:**

There are other directions where trainable activations (not rational ones) are considered. What advantage does the family of rational activations have compared to other ones?

**Limitations:**

Yes

**Strengths And Weaknesses:**

Strengths: The paper is overall well written and the theoretical analysis is sound.
Algebraic activations including rational ones are not explored enough in the literature even though they show promising results. This paper addresses an important aspect of these networks, expressivity, which is essential for the approximation power of neural networks.

Weaknesses:
1) I found the comparison of neural networks with a fixed activation such as GELU with the one using an infinite family of rational activations a bit unfair. Of course expecting these trainable activations to be more expressive is not a surprise and one could also take an infinite family by taking a linear combination of several activations e.g., tanh and GELU.
2) Higher expressivity usually comes at a cost which is higher sample complexity! This paper, as far as I understood, does not mention this aspect.
3) There are important related works that are missing in this paper. The most important one is [1], which discusses the expressivity of rational neural networks from the lens of algebraic geometry. The other related work is [2] which analyses the same family of rational functions (called taylor variety). Polynomial activations are also universal approximators and closely related to rational ones. The expressivity of these networks are examined in several papers  [3,4,5,6,7]. For the analysis of sample complexity of rational networks also [8] is valuable.

[1] Grosdos et al., Algebraic geometry of rational neural networks, 2025.

[2] Conca et al., Taylor Polynomials of Rational Functions, 2023.

[3] Marchetti et al., An Invitation to Neuroalgebraic Geometry, 2025.

[4] Kileel et al., On the Expressive Power of Deep Polynomial Neural Networks, 2019.

[5] Shahverdi et al., Learning on a Razor’s Edge: the Singularity Bias of Polynomial Neural Networks, 2025.

[6] Massarenti et al., The Alexander–Hirschowitz theorem for neurovarieties, 2025.

[7] Shahverdi et al., On the Geometry and Optimization of Polynomial Convolutional Networks, 2024.

[8] Zhang et al., Covering Number of Real Algebraic Varieties and Beyond, 2023.

---

> ### Author Rebuttal · Authors · 2026-03-30
>
> We sincerely thank the reviewer for the careful reading and the positive assessment of the theory.
> On the comparison with fixed activations: this is correct. Our comparisons are intentionally designed this way because we wanted to explore whether allowing a small trainable rational activation family changes approximation efficiency in practice. Our approximation proofs show that the gain from having trainable rational activation functions is much more than one would expect from parameter count alone, and our experiments are meant to test whether some of that advantage is translated into practice. We agree that trainable activations can be introduced in several ways, and we will update the paper to say this more explicitly. Our reason for focusing on trainable rational activations is precisely that the theory here is unusually sharp: rational activations achieve a $\log\log(1/\varepsilon)$ approximation regime where smooth fixed activations such as GELU require $\log(1/\varepsilon)$. This is the key advantage we exploit, and it is not something generic trainable activation families are known to provide.
>
> 1. On sample complexity: this is a very good concern. However, our results are about representational efficiency, not statistical efficiency. The point of the theory is that, for a fixed target accuracy, rational activations can realize the approximation with much fewer effective degrees of freedom in depth/size than smooth fixed activations. So the expressivity gain is not coming from paying for a much larger model class in the way one usually worries about in sample-complexity arguments. We agree that a more complicated hypothesis class can have higher sample complexity in general, but that is not the mechanism behind the improved expressibility shown here. We will make this distinction more explicit in the revision.
>
> 2. On related work: thank you, this is very helpful. We agree that these papers should be discussed. The algebraic-geometry and neuroalgebraic references are relevant and complementary, but they study a rather different notion of expressivity from the one in our paper. Our focus is on quantitative approximation efficiency and explicit separations of the form $\log\log(1/\varepsilon)$ versus $\log(1/\varepsilon),$ whereas those works analyze algebraic, geometric, and structural properties of polynomial/rational network classes. Likewise, the sample-complexity reference is relevant context for a broader discussion of rational networks, even though it is not the main object of our analysis here. We will add and discuss these references in the revision.
>
> 3. On the question “There are other directions where trainable activations (not rational ones) are considered. What advantage does the family of rational activations have compared to other ones?”: the main advantage is that rational activations are one of the very few trainable families for which we can currently prove a strong approximation-efficiency benefit. In particular, the paper shows that rational activations can approximate broad smooth activation families with doubly-logarithmic complexity, while smooth non-rational activations can require logarithmic complexity even to approximate simple rational targets. That qualitative $\log\log$ versus $\log$ gap is the central reason we study this family. So our emphasis is not that rational activations are the only trainable activations one could use, but that they are a mathematically special family for which the efficiency gain can be proved sharply and then tested empirically.
>
> We are very grateful for these thoughtful comments. They will help us improve both the framing and the bibliography of the paper, and we hope the above clarifications address the reviewer’s concerns. We will revise the paper accordingly.

---

> > ### Author Rebuttal · Reviewer_pPPX · 2026-04-02
> >
> > I thank the authors for their rebuttal. I believe that this work would be more complete with the addition of a brief discussion on sample complexity, as well as a clearer positioning within the related literature, particularly the works of [1] and [2].
> >
> > I should also mention that the log-log behavior of rational functions may be connected to the results in [2] from an algebraic perspective, as they consider the family of rational functions. In particular, the expressivity of such families can be studied through invariants such as the "degree" of the so-called Taylor variety, which captures how twisted or space-filling the family is.
> >
> > I will revise my score to weak accept.

---

### Official Review · Reviewer_thbo · 2026-02-21

**Soundness:** 3
**Presentation:** 2
**Significance:** 2
**Originality:** 2
**Overall Recommendation:** 4
**Confidence:** 4

**Summary:**

This paper analyzes the approximation capability of neural networks with a rational polynomial activation function. The authors prove that while rational polynomials can approximate other activation functions using loglog parameters, other activation functions require log parameters.

**Compliance With Llm Reviewing Policy:**

Affirmed.

**Final Justification:**

Most of my concerns regarding the efficiency of the rational activation function have been addressed. Therefore, I will improve the score to weak accept.

**Key Questions For Authors:**

1. Can authors explain the necessity or superiority of loglog parameters?

2. Can authors conduct more experiments?

**Limitations:**

Yes

**Strengths And Weaknesses:**

Strength



The claim of the paper is solid, and I could not find any flaws in the paper although I didn’t check all the details.



Weakness

The importance of the exponential gap between loglog and log parameters is not explained appropriately. A log parameter is already a pretty tiny number. Is it really necessary to reduce it to loglog parameters?

If the authors want to claim that this discrepancy between the approximation rates is important, they should conduct an experiment to validate that the rational polynomial can approximate a target function with much smaller parameters efficiently. For example, authors can change the number of parameters and observe the discrepancy between the accuracy of networks with the rational polynomial and other activation functions.  Currently, I couldn’t find such an experimental result in the paper.

Also, it is preferable to target a function with a kink or a high curvature.

---

> ### Author Rebuttal · Authors · 2026-03-30
>
> We sincerely thank the reviewer for the careful reading and the helpful questions.
>
> 1. Thank you for this excellent point, which we will make more explicit in the revised manuscript. The significance of the $\log\log(1/\varepsilon)^3$ versus $\log(1/\varepsilon)$ gap is that it is exponential in the required accuracy. These are extremely different asymptotic regimes. If a class of functions needs $m\sim \log(1/\varepsilon)$ parameters, then the attainable error behaves like $\varepsilon\sim e^{-cm}$ for some $c$. If instead it needs $m\sim \log\log(1/\varepsilon)^3,$ then the attainable error behaves like $\varepsilon\sim e^{-e^{cm^{⅓}}}.$ Thus, each additional parameter in a rational network can buy something like a doubly-exponential reduction in error, rather than a merely exponential reduction. This is why the asymptotic gap is mathematically substantial even if both expressions may look numerically small at first glance. We will clarify this interpretation in the revision.
>
> 2. We also agree with the reviewer’s suggestion that a direct parameter-scaling experiment is valuable. In fact, the paper already contains one parameter-scaling signal on the vision side through VGG4 versus VGG8: in the plain CIFAR setting, these correspond to about $1{,}001{,}866$ versus $4{,}705{,}866$ trainable parameters, so the paper already changes the parameter budget in a controlled way. We agree, however, that the reviewer’s suggested comparison is important, and following exactly this suggestion we ran an additional IQL MuJoCo width sweep with smaller and larger models. The original RL setup in the paper uses two hidden layers of width $256$; the new runs use $128\times128$ (“less”) and $384\times384$ (“more”) instead. Since rational activations themselves have trainable coefficients, the rational models have slightly more parameters than the GELU models, but the gap is tiny: only $80$ extra parameters in total. To make the comparison completely clear, we report the normalized score, raw return, and total parameter count together below.
>
> | Task               | Size | Act      |  Params | Norm score | Raw return |
> | ------------------ | ---- | -------- | ------: | ---------: | ---------: |
> | HalfCheetah-medium | less | GELU     |  77,967 |      55.76 |   8,934.25 |
> | HalfCheetah-medium | less | Rational |  78,047 |      91.54 |  14,844.79 |
> | HalfCheetah-medium | more | GELU     | 627,087 |      94.16 |  15,278.18 |
> | HalfCheetah-medium | more | Rational | 627,167 |      95.40 |  15,483.55 |
> | Hopper-medium      | less | GELU     |  73,737 |      92.13 |   3,555.79 |
> | Hopper-medium      | less | Rational |  73,817 |      90.15 |   3,479.54 |
> | Hopper-medium      | more | GELU     | 614,409 |      91.85 |   3,545.00 |
> | Hopper-medium      | more | Rational | 614,489 |      92.53 |   3,571.13 |
> | Walker2d-medium    | less | GELU     |  77,967 |      43.23 |   2,961.12 |
> | Walker2d-medium    | less | Rational |  78,047 |      51.41 |   3,521.05 |
> | Walker2d-medium    | more | GELU     | 627,087 |      63.76 |   4,366.69 |
> | Walker2d-medium    | more | Rational | 627,167 |      78.96 |   5,407.05 |
>
> We believe this directly addresses the reviewer’s concern. The key point is that a much smaller rational model can often perform comparably to a much larger one while using far fewer parameters. （except for the hopper medium cases where the base model saturated on the performance already leaving only small room for improvement) This is exactly the parameter-efficiency phenomenon suggested by the review, and the new less/more sweep makes it explicit. We are happy to add this experiment in the revision.
>
> 3. We agree that targeting a function with a kink or high curvature is a natural way to connect experiments more directly to the theory, and we will say this more explicitly. At the same time, our empirical goal is intentionally broader: not only to verify the theory on specially constructed high-curvature targets, but also to test whether the advantage persists on standard learning problems that are not engineered around such structure. In fact, the paper already notes a key curiosity in VGG8: the learned rational activations are evaluated mostly in regions where they are smooth and slowly varying, so they are not being used to approximate kinks or high curvature directly, yet they still improve performance. We view this as an important message rather than a weakness: the theory identifies where the separation is sharpest, while the experiments suggest that the practical benefit can extend beyond those special cases to more general settings. We will revise the discussion to make this distinction much clearer.
>
> We are very grateful for these suggestions. They improve both the clarity and the practical relevance of the paper, and we hope the additional explanation and new experiment address the reviewer’s concerns. We will revise the paper accordingly.

---

> > ### Author Rebuttal · Reviewer_thbo · 2026-04-02
> >
> > Most of the issues I was concerned about have been addressed. Therefore, I will raise the score to 4.

---

### Official Review · Reviewer_5JoQ · 2026-03-09

**Soundness:** 4
**Presentation:** 4
**Significance:** 3
**Originality:** 3
**Overall Recommendation:** 4
**Confidence:** 3

**Summary:**

This paper argues that neural networks with trainable low-degree rational activation functions (ratios of polynomials) are provably more expressive and parameter-efficient than networks built from standard fixed activations such as: ReLU and its variants, GELU, Swish /SiLU, Mish, ELU / SELU / CELU, Softplus, Tanh, Softmax family. The core claim is that rational activations achieve exponentially better approximation efficiency in terms of parameter complexity when approximating certain functions - especially those involving sharp transitions, high curvature, or nearby complex singularities.

**Compliance With Llm Reviewing Policy:**

Affirmed.

**Final Justification:**

I find the paper original and well-presented, with somewhat significant theoretical contributions. I raised concerns during the rebuttal, especially regarding the experimental aspect. These concerns were resolved to an extent. In summary, I still believe that the paper might deserve acceptance, but am not completely convinced.

**Key Questions For Authors:**

- The major advantage of using rational approximations in mathematics is related to the Pade' approximation method. Is it implicitly used in your approach? If yes, then one should probably mention this explicitly.

- You use rational functions which do not have poles in the required domain. In general, this property is not preserved under superposition of functions. How does the software deal with that?

**Limitations:**

Yes

**Strengths And Weaknesses:**

## Strengths

The paper argues that rational activations are  theoretically superior in approximation efficiency and often empirically stronger under identical architectures and suggests that rational activations may form a mathematically natural and computationally efficient foundation for future neural network design.

More precisely,  the paper claims the following.

- **Rational networks approximate smooth activations efficiently**. For example, for smooth activations such as GELU, rational networks can approximate them with size $O(\log^3 \log(1/\epsilon))$ . The  obtained lower bound is $O(\log \log(1/ \epsilon)) $. This is  doubly logarithmic in error. These results extend to: ELU, SiLU, Mish, Softplus, Tanh, Softmax family, LogSigmoid, etc. The proof uses classical rational approximation theory and decomposition of GELU into elementary functions (sqrt, log, tanh, etc.), each approximated by rational subnetworks with double-exponential convergence.


- **Smooth networks approximate rationals inefficiently**. There exist simple rational functions such that any bounded-parameter smooth-activation network approximating them to error $\epsilon$ must have size $O(\log(1/\epsilon)) $, which is exponentially worse than the rational case. The argument relies on curvature constraints, nearby poles in rational functions, growth limits of derivatives in smooth networks. Therefore, one has an exponential separation in efficiency  between using smooth and rational activation functions.

I find the paper well-written and concise. It includes many references to the recent literature on rational activation functions and some references to approximation theory. The proofs of the main results look sound.

## Weaknesses

Although the authors appeal to the general advantages of rational approximations compared to the smooth approximations, I do not see any serious use of the existing  theory of rational approximations in their theoretical considerations (except for very special cases of elementary functions). See my question below about this.

Moreover, I believe that the experimental part has a number of shortcomings:

- The experiments are on a rather small scale, with limited hardware, and deploy an existing implementation of rational functions. While this is mentioned by the authors (Section 5), I believe it is insufficient to draw serious conclusions about substantial usefulness of rational activation functions;


- The presented score differences show only quite small efficiency gains for rational activation functions compared to the standard architecture;


- The authors claim that rational activation functions are most useful in cases of singularities and high curvature. However, they mention in the VGG8 experiments that hidden-layer activations evaluate inputs mostly in regions where the functions are smooth and slowly varying. This makes me wonder (unless further clarified) about the usefulness of this approach.

---

> ### Author Rebuttal · Authors · 2026-03-30
>
> We sincerely thank the reviewer for the careful reading and strong positive assessment.
>
> 1. On Padé: while Padé approximation is a classical topic in rational approximation theory, it is not the mechanism behind the main results of this paper. Padé is relevant historical context, but it is not the mechanism driving our main theorems. Padé appears only as a practical connection to prior learnable rational-activation parameterizations and initializations. The theory is driven by different tools: the upper bounds come from constructive rational-network approximation results, while the lower bounds come from nearby-pole / high-curvature obstructions. In particular, nearby poles force local curvature, more precisely second derivatives of order $\varepsilon^{-1},$ while bounded-parameter smooth-activation networks have controlled derivative growth, yielding the $\Omega(\log(1/\varepsilon))$ barrier. We will revise the introduction so that this mechanism is completely clear.
>
> 2. Regarding poles under composition, the software does not depend on accidental cancellations. In the theoretical model we require the denominator to stay nonzero on the real domain of interest. In the experiments, the implementation enforces a strictly positive denominator at every layer, so real poles are excluded by construction. Therefore each activation is well-defined on real inputs, and compositions of such activations remain well-defined as well. We agree this point should be stated more explicitly, since it directly addresses the reviewer’s concern about practical validity under superposition.
>
> 3. We very much appreciate the concern about experimental scale. Our intention is not for the empirical section to settle the final large-scale picture, but to provide meaningful evidence at the scale where a theory paper can already be informative. In our view, small-to-mid scale experiments are useful here because they test whether the proposed advantage is only approximation-theoretic or whether it already appears in practice across representative modern settings. Just as importantly, one of our motivations is to encourage more work in this direction: our hope is that a mathematically grounded theory, together with consistent evidence at this scale, can motivate broader future studies. We consider that an important next step, and it is a direction we fully intend to pursue when broader compute resources are available.
>
> 4. The gains are more meaningful than they may first appear. When a baseline is already very strong, even a smaller absolute gain can still be significant; when the task is less saturated and traditional activations do not perform as well, there is more headroom, so the same activation advantage can translate into a much larger gain. This is exactly the pattern we observe. In offline RL, where the tasks are harder and much less saturated, the gains become large again, e.g. IQL HalfCheetah 94.71 vs. best fixed 85.78 and TD3+BC Walker2d 84.65 vs. best fixed 69.02. As discussed in our response to Reviewer 3, this advantage becomes even clearer when the model size is reduced. Our view is therefore that rational activations are especially valuable in harder, less-saturated regimes, while still being a safe replacement when the baseline is already strong.
>
> 5. Finally, on the connection between theory and practice: our intention was not to claim that rational activations are useful only when hidden activations literally operate near singularities or in highly nonsmooth regions. The high-curvature discussion identifies where the sharpest asymptotic separation occurs; it is a worst-case approximation-theoretic explanation, not a claim that all practical gains must come from that extreme regime. In practice, the benefit can be broader. As the reviewer notes, in VGG8 many hidden-layer evaluations lie in regions where the learned rational is smooth and slowly varying. We view this as evidence that the practical advantage comes from adaptive local shaping: the learned rational can match the layerwise input distribution and place curvature where it is most useful, even when the network is not operating directly at a singular point. We will revise this discussion so the paper more clearly separates the sharp theoretical regime from the broader adaptive-shaping mechanism suggested by the experiments.
>
> We are very grateful for these comments. They will help us make the paper clearer and stronger, and we hope the above clarifications address the reviewer’s concerns. We will revise the paper accordingly.

---

> > ### Author Rebuttal · Reviewer_5JoQ · 2026-04-02
> >
> > I thank the authors for their reply. My concerns have been mostly resolved. I will keep my (somewhat positive) score.

---

### Official Review · Reviewer_idk1 · 2026-03-12

**Soundness:** 3
**Presentation:** 3
**Significance:** 3
**Originality:** 3
**Overall Recommendation:** 4
**Confidence:** 3

**Summary:**

This paper studies neural networks with trainable rational activation functions, and argues that they have both theoretical advantages over standard fixed activations. The main theoretical result in this paper shows that rational functions can approximate GELU efficiently on a compact domain, while approximating rational function with GELU requires greater complexity in the worst case. This paper then generate the results to full neural networks and discuss extensions to gating and normalization. This paper then demonstrates in multiple tasks that replacing standard activation with (low-degree trainable) rational neural network will result in better performance.

**Compliance With Llm Reviewing Policy:**

Affirmed.

**Key Questions For Authors:**

1. As stated in Weakness 1, the current construction of $R(x)$ in Theorem 3.2 depends on $\epsilon$, then does there exist a rational function $R(x)$ that doesn't depend on $\epsilon$ for Theorem 3.2 to work? Furthermore, will Theorem 3.2 still work without uniformly bounding parameters of the GELU network?

2. In Theorem 3.1 and 3.3, results are shown for the size of networks (rational network or GELU-activated network). As "size" is determined by both depth $M$ and width $L$, is $M$ fixed when the "size" of the constructed network increases with $\epsilon$ or that $M$ is changing with $\epsilon$ to get the desired approximation rate? If $M$ is changing with $\epsilon$, what would be the order of $M$ for the constructed network?

3. In Equation (2), it is said that $r^{(l)}$, the rational activation at the $l$-th layer, potentially have its own coefficients. In the construction of the rational network in Theorem 3.1, is the degree ($r_P, r_Q$) of $r^{(l)}$ uniformly bounded for all $l$?


4. In Section 5, the paper attributes part of the increased wall-time to framework and kernel-launch overhead. However, since the rational activation is itself trainable, the overhead is not merely that of evaluating a more complicated fixed function: backpropagation must also update the activation’s internal parameters. Thus, at least part of the runtime increase seems intrinsic to the use of adaptive rational activations rather than something that can be eliminated by better parallelization or native implementation alone. Could the authors clarify roughly what fraction of the observed wall-time increase is due to this intrinsic reason?

**Strengths And Weaknesses:**

Strength:

1. This paper provides upper bound and lower bound on both approximation directions, giving a clear characterization of the approximation gap.

2. The main theoretical comparison is made against GELU, which is more relevant to current architectures than the comparison between ReLU and rational network in the existing literature.

3. The discussion of the interaction between adaptive rational activations and normalization layers offers a new perspective on improving performance.

4. The paper provided experiments on multiple tasks to demonstrate that trainable rational activation can improve performance in practice.


Weakness:

1. In the statement of Theorem 3.2, a natural understanding of this theorem would be that there exists a rational function $R$ such that $\forall \epsilon >0 $, a GELU network that $\epsilon$-approximates $R$ requires $\Omega (log(1/\epsilon))$ size. However, as shown in the proof of theorem B.2., the construction of $R(x) = 1/(x^2+ \eta^2)$ with $\eta < \epsilon^{1/4}$ depends on $\epsilon$, which contradicts the natural understanding. Maybe improve the theorem statement to make it clear.

2. In table 5 "ViT S 8 no norm" case, models using ReLU and GELU activations are marked "failed", with explanation of "fail to train to a usable solution", which is a little coarse. It would improve the experimental clarity if the appendix provided more detail on the failure details (e.g. NaNs, unstable optimization, or collapse to very poor accuracy), together with representative training curves or summary statistics across seeds.

---

> ### Author Rebuttal · Authors · 2026-03-30
>
> We sincerely thank the reviewer for the careful reading, positive assessment, and very helpful questions.
>
> 1. You are right. What our proof establishes is a worst-case family rather than the same lower bound for one fixed rational $R$ independent of $\varepsilon$. We used the Appendix-B family $r_\eta(x)=\frac{1}{x^2+\eta^2}$ because it cleanly witnesses the full worst-case $\Omega(\log(1/\varepsilon))$ barrier in Theorem 3.2. The common mechanism, however, is not the specific formula, but the need to reproduce very large curvature on a small scale, more precisely second derivatives of order $\varepsilon^{-1}$. This is the feature shared by the worst-case nearby-pole family and by fixed non-rational high-curvature examples such as $|x|$: both force smooth bounded-parameter networks to generate rapidly growing local curvature, while rational networks can realize such behavior much more efficiently. The difference is quantitative. For fixed targets such as $|x|$, the separation is the familiar log-vs-loglog regime, i.e. a $\log(1/\varepsilon)/\log\log(1/\varepsilon)$-type overhead rather than the full worst-case logarithmic overhead. The nearby-pole family is chosen because it upgrades this same large-curvature mechanism to the full worst-case logarithmic barrier, which is exactly what Theorem 3.2 requires. We will revise the discussion to make this hierarchy explicit: fixed high-curvature targets already show a strict gap, while the $\varepsilon$-dependent nearby-pole family is the clean witness for the sharp worst-case statement. We will also clarify that the present lower-bound proof uses the uniform parameter bound essentially, since that bound is what controls the derivative growth available to the smooth-activation network.
>
> 2. In Theorems 3.1 and 3.3, “size” means the number of trainable parameters. In the actual constructions, the width remains O(1), so the $\varepsilon$-dependence is carried by the depth M. Thus M is not fixed; it grows with $\varepsilon$, and because width is constant, M has the same order as the size up to constants. Concretely, Theorem 3.1 gives constant-width rational networks with size/depth $O((\log\log(1/\varepsilon))^3)$, while Theorem 3.3 gives constant-width GELU networks with size/depth $O((\log(1/\varepsilon))^2)$. We appreciate this question and will make this point fully explicit in the revision so the role of depth versus width is completely unambiguous.
>
> 3. Yes: the rational degrees are uniformly bounded across layers. Equation (2) allows different coefficients in different layers, but the constructions keep the numerator and denominator degrees fixed and low throughout. The expressive advantage comes from composition across depth, not from increasing the degree with layer index or with $\varepsilon$. We will make this more explicit around Eq. (2) and in the theorem discussion.
>
> 4. We agree that Table 5 should be stated more clearly. In the no-LayerNorm ViT-S/8 runs, ReLU and GELU did not merely underperform; they diverged and produced no usable checkpoint all the time from the start. We will revise “failed” to a precise statement such as “diverged to NaNs / no usable checkpoint under the shared no-norm setup.” More importantly, this behavior is not surprising under the unchanged training recipe: standard Transformers rely on LayerNorm as a basic stabilizing component, so removing it while keeping a fixed activation leaves no adaptive mechanism to absorb the resulting instability. The rational case is different because the activation itself is trainable and can adapt its local scaling and shape after normalization is removed. We believe making this point explicit will significantly improve the presentation of Table 5.
>
> 5. We very much appreciate this runtime question, and we agree that the intrinsic cost of adaptivity should be separated from generic implementation overhead. We therefore measured this directly by comparing matched training steps for two otherwise identical rational models: one with trainable rational coefficients and one with those coefficients frozen, timing only forward/backward/optimizer compute. On VGG8/CIFAR-10, 10 matched steps took $0.22005\text{s}$ in the trainable case versus $0.21862\text{s}$ in the frozen case, so the extra cost attributable specifically to learning the rational coefficients was $0.00143\text{s}$ over 10 steps, i.e. about $0.143\text{ ms/step}$ and about 0.65% of train-step compute. Extrapolated over the full run, this is about $3.35\text{s}$ versus $524.20\text{s}$ total train compute time. Thus, the adaptive-parameter update itself is only a very small part of the runtime increase; most of the gap comes from the current unfused implementation rather than from intrinsic adaptivity. We will add this clarification in the revision.
>
> We are extremely grateful for your thoughtful comments.

---

> > ### Author Rebuttal · Reviewer_idk1 · 2026-04-03
> >
> > My questions regarding some theoretical parts are answered and resolved. I will maintain my score.

---

### Decision · Program_Chairs · 2026-04-30

**Decision:**

Accept (regular)

**Comment:**

The reviewers agree that the paper is well written and provides interesting results. Specifically, the exponential separation gap between how networks with rational activations approximate networks with smooth activations, compared to the other way around. The experimental section complements these theoretical results, although some of the reviewers found it rather low-scale. I believe this limitation is acceptable given the mainly theoretical contribution.

I suggest that the authors incorporate the reviewer's comments for the camera-ready version.